# ARTIFICIAL HIPPOCAMPUS NETWORKS FOR EFFICIENT LONG-CONTEXT MODELING

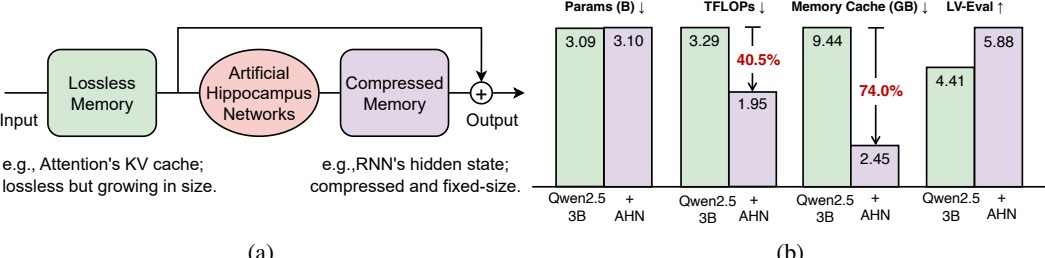

Figure 1: (a) Artificial Hippocampus Networks (AHNs) transform lossless memory into fixed-size compressed representations for efficient long-context modeling. Lossless memory (e.g., attention's KV cache) stores exact input information but grows with sequence length, leading to high cost for long sequences. In contrast, compressed memory (e.g., RNNs' hidden state) maintains a constant cache size and computational cost per input token, but inevitably loses details. In our framework, a sliding window attention maintains exact recent context as lossless short-term memory, while AHN recurrently compresses out-of-window information into a fixed-size state as compressed long-term memory. This allows the model to process long sequences efficiently, retaining both precise short-term information and a compact summary of history. See Figure 2 for more details. (b) On the long-context benchmark LV-Eval (128k sequence length), augmenting Qwen2.5-3B-Instruct with AHNs (+0.4% parameters) reduces FLOPs by 40.5% and memory cache by 74.0%, while improving average score from 4.41 to 5.88.

## ABSTRACT

Long-sequence modeling faces a fundamental trade-off between the efficiency of compressive fixed-size memory in RNN-like models and the fidelity of lossless growing memory in attention-based Transformers. Inspired by the Multi-Store Model in cognitive science, we introduce a memory framework of artificial neural networks. Our method maintains a sliding window of the Transformer's KV cache as lossless short-term memory, while a learnable module termed Artificial Hippocampus Network (AHN) recurrently compresses out-of-window information into a fixed-size compact long-term memory. To validate this framework, we instantiate AHNs using modern RNN-like architectures, including Mamba2, DeltaNet, and GatedDeltaNet to augment open-weight base LLMs. We also propose an efficient self-distillation method where the base model's all parameters are frozen and only the parameters from AHNs are optimized. For inference, our method sets a default large sliding window size of 32k for attention, and AHNs activate only when the sequence length exceeds the 32k window, addressing the quadratic-complexity issue of attention that emerges at that scale. Extensive experiments on long-context benchmarks LV-Eval and InfiniteBench demonstrate that AHN-augmented models consistently outperform sliding window baselines and achieve performance comparable or even superior to full-attention models, while substantially reducing computational and memory requirements. For instance, augmenting the Qwen2.5-3B-Instruct with AHNs reduces inference FLOPs by 40.5% and memory cache by 74.0%, while improving its average score on LV-Eval (128k sequence length) from 4.41 to 5.88. Code and models will be released to facilitate future research.

# 1 INSTRUCTION

*"Memory is the treasury and guardian of all things"* (Cicero, 55 BCE). Inspired by the fundamental role of memory in intelligence, researchers have long sought to model this cognitive function in artificial systems. Early efforts centered on Recurrent Neural Networks (RNNs) (Werbos, 1988; Jordan, 1986; Elman, 1990; Hochreiter & Schmidhuber, 1997; Cho et al., 2014; Hopfield, 1982; 1984), where sequential information is encoded by continuously updated hidden states. Over time, diverse paradigms for memory representation emerged, including key-value (KV) caches in attention mechanisms (Vaswani et al., 2017), external memory modules in Neural Turing Machines and Memory Networks (Graves et al., 2014; Weston et al., 2015), and external databases for retrieval-augmented models (Lewis et al., 2020). Among these, RNN-like and attention-based models have become the most widely used, each offering distinct advantages and limitations (Yu & Wang, 2025; Lieber et al., 2024).

RNN-like models compress all historical information into a fixed-size hidden state, which can be treated as memory. At each step, they update the memory using the current input and the previous memory. This design ensures constant memory and computation per step, making them efficient for long sequences. However, compressing all information into a fixed-size memory inevitably leads to information loss, especially in tasks that require precise long-range information recall (Wen et al., 2025).

To address the limitations of RNNs, attention mechanisms and the Transformer architecture are introduced (Bahdanau et al., 2015; Luong et al., 2015; Vaswani et al., 2017). In causal attention, the key-value cache functions as memory: for each input token, a new key and value are generated and appended to the cache. Unlike RNNs, this memory is essentially lossless, as it retains all token-level information, thereby providing much higher memory capacity. The introduction of the Transformer quickly revolutionized sequence modeling, giving rise to a series of powerful models (Radford et al., 2018; 2019; Devlin et al., 2019; Brown et al., 2020; OpenAI, 2023). Yet, the lossless nature of KV cache is a double-edged sword: while it enables powerful memory retention, the memory size grows linearly with sequence length, and the total computational cost of attention updates scales quadratically. This becomes a significant challenge when processing extremely long sequences.

When Transformers with growing lossless memory struggle for very long sequences, it is natural to revisit the RNNs' fixed-size compressed memory, which offers constant per-token update cost regardless of context length (Katharopoulos et al., 2020; Gu & Dao, 2024; Yang et al., 2024c). This contrast highlights a fundamental trade-off between the efficiency of compressive memory and the fidelity of lossless memory. To address this problem, it is instructive to consider how the human brain maintains nearly constant volume through early and middle adulthood (Dekaban & Sadowsky, 1978; Courchesne et al., 2000; Fotenos et al., 2005) while still supporting efficient processing of information across the human lifespan. The theory of Multi-Store Model of memory (MSM) in Cognitive Science and Neuroscience (Atkinson & Shiffrin, 1968) suggests that although lossless short-term memory (or called working memory (Baddeley & Hitch, 1974)) has limited capacity and duration (Miller, 1956; Atkinson & Shiffrin, 1968; Peterson, 1959), the hippocampus continually consolidates them into long-term cortical representations (Scoville & Milner, 1957; Squire & Zola-Morgan, 1991; Alvarez & Squire, 1994; McClelland et al., 1995; Eichenbaum, 2000; Takashima et al., 2009).

Inspired by MSM (Atkinson & Shiffrin, 1968), we propose an artificial neural memory framework that converts lossless short-term memory into compressed long-term memory. Our method maintains a sliding window of the Transformer's KV cache as lossless short-term memory. Information that moves beyond this window is processed by a learnable compression module we term the Artificial Hippocampus Network (AHN). This network recurrently compresses the out-of-window context into a fixed-size state as the long-term compressed memory. AHNs can be instantiated with RNN-like architectures, and the overall framework is illustrated in Figure 1a.

To evaluate the effectiveness of AHNs, we instantiate them using Mamba2 (Dao & Gu, 2024), DeltaNet (DN) (Schlag et al., 2021; Yang et al., 2024d) and GatedDeltaNet (GDN) (Yang et al., 2025), resulting in the AHN-Mamba2, AHN-DN and AHN-GDN. We introduce an efficient self-distillation training scheme in which the teacher model is an open-weight attention-based model (*e.g.*, Qwen), and the student model shares the teacher's parameters but with token mixer of window attention and AHN. We employ a KL divergence loss, optimizing only the AHN parameters while

freezing all remaining parameters, as shown in Figure 2b. The models on trained on ChatQA 2 (Xu et al., 2025) with 1B tokens, sample sequence length up to 24k, and random sliding window size up to 8k, which only cost $\sim 10$ hours on 32 A100 GPUs to train AHNs to augment 7B model. Notably, for inference, we set a default sliding-window attention size of 32k, which is substantially larger than those used in prior attention–RNN hybrid methods (e.g., 64 in (Zhang et al., 2025; Irie et al., 2025)) AHNs activate only when the sequence length exceeds the 32k window, addressing the quadratic-complexity issue of attention that emerges at that scale.

Experimental results on long-context benchmarks LV-Eval (Yuan et al., 2024) and InfiniteBench (Zhang et al., 2024a) show that AHN-augmented models consistently outperform their sliding window counterparts, and match or even surpass full attention models while significantly reducing computational and memory cache costs. For instance, as shown in Figure 1b, augmenting Qwen2.5-3B-Instruct (Yang et al., 2024a) with AHNs (+0.4% parameters) reduces FLOPs by 40.5% and memory cache by 74.0%, while improving average score from 4.41 to 5.88 on LV-Eval (128k sequence length) (Yuan et al., 2024).

The contributions of this paper are twofold. First, we introduce the concept of Artificial Hippocampus Networks (AHNs), which continually transform lossless memory outside the sliding window into a compressed memory representation, enabling the model to leverage both memories for efficient long-context modeling. Second, to empirically validate the effectiveness of AHNs, we instantiate the concept into AHN-Mamba2, AHN-DN, and AHN-GDN, and train these instances using an efficient self-distillation scheme. Experimental results demonstrate that these instances substantially enhance model efficiency on long-sequence benchmarks, while achieving competitive performance compared to the full attention model.

## 2 METHOD

### 2.1 PRELIMINARY

Most modern autoregressive large language models are built on Transformer architecture (Vaswani et al., 2017), which employs self-attention as the core mechanism for token mixing. Given an input sequence of $L$ tokens $X = (x_1, x_2, ..., x_L) \in \mathbb{R}^{L \times D}$ ($D$ is the hidden dimension), self-attention first projects the tokens into query ($Q$), key ($K$), and value ($V$) matrices via learned linear transformations:

$$Q = XW_Q, \qquad K = XW_K, \qquad V = XW_V \tag{1}$$

where $W_Q$, $W_K$, and $W_V$ are trainable weight matrices. The attention output is then computed as a weighted sum of the value vectors:

$$\text{Attention}(Q, K, V) = \text{softmax}\left(\frac{QK^T}{\sqrt{d_{in}}} \odot \mathcal{M}\right) V \tag{2}$$

where $\mathcal{M} \in \mathbb{R}^{L \times L}$ is the causal mask, defined by $\mathcal{M}_{ij} = 1$ if $j \leq i$, and $\mathcal{M}_{ij} = 0$ otherwise.

### 2.2 ARTIFICIAL HIPPOCAMPUS NETWORKS

**Definition.** Inspired by MSM (Atkinson & Shiffrin, 1968) and the hippocampus (Scoville & Milner, 1957) that consolidates lossless short-term memory into compact and long-term representations, we introduce Artificial Hippocampus Networks (AHNs) to emulate this biological function by compressing historical information into a fixed-size recurrent state. An AHN operates alongside a sliding attention window of size $W$. For the token at step $t > W$, the AHN updates the compressive memory by processing the key-value (KV) pair $(k_{t-W}, v_{t-W})$ that just exited the sliding window. This recurrent memory update is defined as:

$$h_{t-W} = \text{AHN}((k_{t-W}, v_{t-W}), h_{t-W-1}) \tag{3}$$

where $h_{t-W}$ is the updated compressed memory summarizing context up to and including position $t - W$. $h_{t-W}$ can be a vector or matrix. Due to the recurrent formulation of Equation 3, AHNs can be implemented with RNN-like architectures, enabling the learnable and efficient compression of long context history.

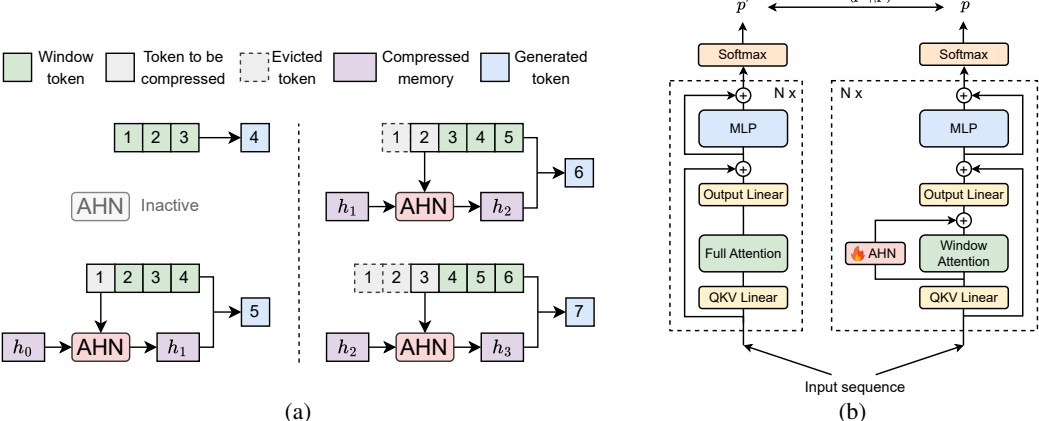

(a)            (b)

Figure 2: (a) Illustration of the model augmented with Artificial Hippocampus Networks (AHNs). In this illustrative example, we set the sliding window length to 3 for clarity. For model inference in our experiments, the default window length is 32k. When the input sequence length is less than or equal to the window length, the model operates identically to a standard Transformer. For longer sequences, AHNs continually compress the token outside the window into a compact memory representation. The model then utilizes both the lossless information within window, and the compressed memory to generate the next token. (b) Self-distillation training framework of AHNs based on an open-weight LLM. During training, the base LLM's weights are frozen, and only the AHNs' parameters are trained.

**Integration with lossless memory.** Within the predefined sliding window, standard causal attention is applied to preserve lossless memory of recent tokens. Once the input sequence length exceeds the window size, AHNs are activated to compress the KV pair outside the window, *i.e.*, $(k_{t-W}, v_{t-W})$, into a fixed-size compressed memory $h_{t-W}$. After this compression, the original KV pair beyond the window can be safely discarded, retaining only the KV cache within the window $\{(k_i, v_i)\}_{i=t-W+1}^t$. Finally, the current query $q_t$ accesses information from both compressed and lossless memories to produce the output:

$$y_t = f(h_{t-W}, \{(k_i, v_i)\}_{i=t-W+1}^t, q_t) \tag{4}$$

An illustration of the overall model mechanism with AHNs is provided in Figure 2a. Besides, the illustration of AHNs with attention sinks (Xiao et al., 2024c) is shown in Figure 6 in the appendix.

### 2.3 INSTANTIATION

As discussed above, AHNs can be instantiated using RNN-like architectures. In our experiments, we focus on modern linear recurrent models for their efficient parallel training. Specifically, we utilize three architectures including Mamba2 (Dao & Gu, 2024), DeltaNet (DN) (Schlag et al., 2021; Yang et al., 2024d), and its enhanced version, GatedDeltaNet (GDN) (Yang et al., 2024c), to instantiate AHNs into AHN-Mamba2, AHN-DN and AHN-GDN, respectively. Below, we present the implementation of AHN-GDN for each head as a representative example, and the other two AHN instances are described in Appendix A.1. Specifically, AHN-GDN updates memory via the gated delta rule (Schlag et al., 2021; Yang et al., 2024d;c):

$$
\begin{aligned}
h_{t-W} &= \text{AHN-GDN}((k_{t-W}, v_{t-W}), h_{t-W-1}, x_{t-W}) \\
&= \alpha(x_{t-W})(\mathbf{I} - \beta(x_{t-W})k_{t-W}^T k_{t-W})h_{t-W-1} + \beta(x_{t-W})k_{t-W}^T v_{t-W}
\end{aligned} \tag{5}
$$

where learnable parameters for per head are $W_\alpha \in \mathbb{R}^{D \times 1}$ in $\alpha(\cdot)$ and $W_\beta \in \mathbb{R}^{D \times 1}$ in $\beta(\cdot)$. Unlike GatedDeltaNet (Yang et al., 2025), which compresses all past tokens, AHN-GDN only compresses tokens outside the sliding window. For each position $t$, the query $q_t$ derived from $x_t$ is used to access the compressed memory $h_{t-W}$. Note that AHNs do not introduce separate QKV projection layers. Instead, they directly transform the lossless memory (*i.e.*, the KV cache) from attention into a fixed-size compact memory. The compressed memory $h_{t-W}$ is further modulated by a gate function $\gamma(x_t)$ and then is transformed by a linear projection to generate output:

$$y_{\text{AHN},t} = \gamma(x_t)q_t h_{t-W} W_o \tag{6}$$

Different from GatedDeltaNet (Yang et al., 2025), the output of $\gamma(x_t) = x_t W_\gamma$ is a scalar for each head with learnable parameter $W_\gamma \in \mathbb{R}^{D \times 1}$, and the output linear is grouped by heads (Krizhevsky

Table 1: Complexity of causal attention with and without AHN-GDN. Here, $L$: input sequence length; $D$: hidden dimension; $N_\text{q}/N_\text{kv}$: number of query/key-value heads; $H$: head dimension; $W$: sliding window size. AHNs are activated only when $L > W$. FLOPs account for matrix multiplication only; softmax, normalization, and matrix element summation are omitted. Items shown in gray can be further omitted compared to the other terms.

| Token mixer | Causal attention (Full) | Causal attention (Window) + AHN-GDN |
|---|---|---|
| Parameters | $2DH(N_\text{q} + N_\text{kv})$ | $2DH(N_\text{q} + N_\text{kv}) + 3DN_\text{q} + H^2 N_\text{q}$ |
| Memory cache | $2LHN_\text{kv} \sim O(L)$ | $2WHN_\text{kv} + H^2 N_\text{q} \sim O(W)$ |
| FLOPs | $4LDH(N_\text{q} + N_\text{kv}) + 2HN_\text{q}L^2 \sim O(L^2)$ | $4LDH(N_\text{q} + N_\text{kv}) + 2HN_\text{q}W^2 + 2(L - W)\times$ $(2WHN_\text{q} + H^2 N_\text{q} + 3DN_\text{q} + H^2 N_\text{q}) \sim O(WL)$ |

et al., 2012; Jiang et al., 2020) with learnable weight $W_o \in \mathbb{R}^{H \times H}$ ($H$ denotes head dimension). Finally, we simply sum the outputs from AHN and the attention mechanism:

$$y_t = y_{\text{AHN},t} + \text{Attention}(\{(k_i, v_i)\}_{i=t-W+1}^t, q_t) \tag{7}$$

**Complexity analysis.** Table 1 summarizes the computational and memory complexities of the attention token mixer with and without AHN-GDN, and Figure 3 compares the complexities of Qwen2.5-3B with and without AHN-GDN. As shown, integrating AHNs significantly improves efficiency over standard full attention in both memory usage and FLOPs. In particular, AHN-GDN reduces the computational complexity of attention to linear in sequence length while keeping the memory cache size constant. By contrast, vanilla full attention incurs quadratic computational cost and memory usage that grows linearly with sequence length.

## 2.4 TRAINING FRAMEWORK

While an AHN-augmented model can be trained from scratch, we adopt a more computationally efficient approach using self-distillation (Hinton et al., 2015; Zhang et al., 2018; 2019). This allows us to leverage powerful pre-trained models. Our training framework uses an open-weight LLM (*e.g.*, Qwen (Yang et al., 2024a)) as the teacher model, with its output probability denoted as $p'$. The student model is the same LLM, but we modify its attention mechanism to operate over a limited receptive field of a sliding window at every layer. These window attention layers are then augmented with AHNs. The student's output probability is denoted as $p$. We train the student to mimic the teacher's output distribution by minimizing the Kullback-Leibler (KL) divergence: $l = \text{KL}(p'\|p)$. To maximize efficiency, the base model's weights are frozen during training, and only the AHN parameters are optimized. Taking AHN-GDN as an example, only the parameters involved in Equations 5 and 6 are learnable. For each attention head, these trainable parameters consist of the gating weights $W_\alpha \in \mathbb{R}^{D \times 1}$ in $\alpha(\cdot)$, $W_\beta \in \mathbb{R}^{D \times 1}$ in $\beta(\cdot)$, $W_\gamma \in \mathbb{R}^{D \times 1}$ in $\gamma(\cdot)$ as well as the output projection $W_o \in \mathbb{R}^{H \times H}$. Here, $D$ and $H$ denote the hidden dimension and the head dimension, respectively. With $N_\text{q}$ attention heads, the model contains $N_\text{q}$ such sets of parameters, amounting to only $\sim 0.4\%$ relative to the frozen base model's parameters. The framework is illustrated in Figure 2b.

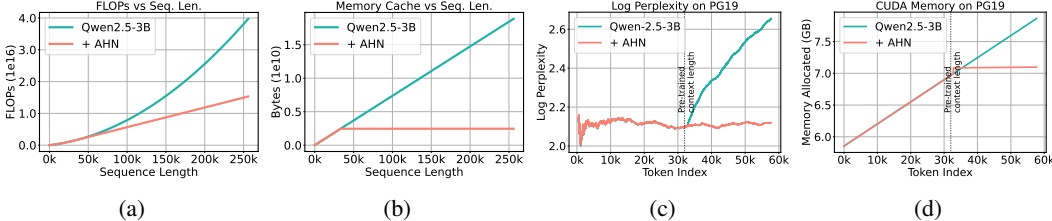

(a)  (b)  (c)  (d)

Figure 3: Complexity analysis of the Qwen2.5-3B-Instruct and model perplexity, with and without AHNs. AHNs are only activated when the sequence length exceeds the window size (32k in this example). (a) The model with AHN enjoys linear computational complexity with respect to sequence length. (b) The model with AHN maintains a consistent memory cache size. (c) Perplexity results on the first book of the PG19 test set (57K tokens). While Qwen-3B-Instruct degrades beyond its pre-trained context length, AHN-augmented models maintain consistently low perplexity. (d) Peak GPU memory under the same example.

## 3 EXPERIMENTS

### 3.1 SETUPS

**Models and datasets.** We build our AHNs on top of open-weight Qwen2.5-Instruct series (3B, 7B, 14B) (Yang et al., 2024a). To demonstrate architectural flexibility, we implement the AHN module using three modern recurrent models: Mamba2 (Dao & Gu, 2024), DeltaNet (Schlag et al., 2021; Yang et al., 2024d), and GatedDeltaNet (Yang et al., 2024c). The training data is ChatQA2 dataset (Xu et al., 2025), an open-source collection of diverse long-context tasks. We evaluate our methods across a comprehensive suite of long-context benchmarks, including LongBench (Bai et al., 2024a), InfiniteBench (Zhang et al., 2024a), and LV-Eval (Yuan et al., 2024), with an additional illustrative example drawn from PG19 (Rae et al., 2020).

**Baselines.** We evaluate AHN-augmented models against two primary baselines: sliding window attention (SWA) with attention sinks (Xiao et al., 2024c) and the Compressive Transformers (CT) (Rae et al., 2020). We implement the Compressive Transformer using max and average pooling to compress tokens outside the sliding window at a $4\times$ compression rate. To ensure a fair comparison, all methods are allocated the same lossless memory budget, and the memory size of compressed tokens for CT is set to equal the memory size of the hidden state of AHNs. The performance of full attention is also reported as a reference.

**Implementation details.** We implement all AHN instances in PyTorch (Paszke et al., 2019), building on LLaMA-Factory (Zheng et al., 2024) and Flash Linear Attention (Yang & Zhang, 2024). During training, we freeze the entire base LLM and only train the newly added AHN module (only $\sim 0.4\%$ parameters relative to base LLM) using self-distillation, as illustrated in Figure 2b. To ensure the AHN module learns a generalizable compression strategy, we randomize the starting position (the number of attention sinks) of the AHN modules and also the sliding window size. Specifically, we use a maximum sequence length of 24k tokens during training. For each example, the attention-sink size is uniformly sampled from [0, 32, 64, 128, 512, 2048, 4096] after removing any candidates larger than half of the sequence length. The total token number of lossless memory (attention sinks + sliding window) is uniformly sampled from [32, 64, 128, 256, 512, 1024, 2048, 4096, 8192] after filtering out values smaller than one-eighth of the sequence length. For optimization, we use the AdamW (Loshchilov & Hutter, 2019) optimizer with a learning rate of 1e-4, which is warmed up linearly over the first 10% of steps and then cosine decayed. All models are trained for one epoch on the ChatQA2 dataset, consisting of 1B tokens, using a global batch size of 128 for a total of 740 update steps. Training AHNs for a 7B base model requires only $\sim$10 hours on 32 A100 GPUs.

### 3.2 AN ILLUSTRATIVE EXAMPLE

By compressing historical information beyond the sliding window into a fixed-size memory, AHN-augmented models significantly reduce both computational complexity and memory footprint, as shown in Figure 3a and 3b. We demonstrate this advantage with a real example on a 57K token passage from the PG19, a benchmark of long-form books designed to test extended context understanding. We compare the base 3B-Instruct models against their AHN-GDN counterparts. As shown in Figure 3c, the perplexity of standard Qwen models rises sharply once the 32k token context window is exceeded. In contrast, the AHN-GDN augmented model maintains consistently low perplexity. Furthermore, Figure 3d illustrates that while the base models' memory usage grows linearly under FlashAttention, AHN-GDN keeps the CUDA memory usage nearly constant, highlighting its effectiveness for processing long-context sequences.

### 3.3 LONG-CONTEXT BENCHMARKS

We now systematically evaluate AHN-augmented models on long-context benchmarks to assess their effectiveness and efficiency. Our evaluation is structured across two settings: First, we conduct ultra-long-context evaluation on InfiniteBench (Zhang et al., 2024a) and LV-Eval (Yuan et al., 2024) (both use 128k-length subset), comparing AHN-augmented models with full attention, sliding window attention (SWA) with attention sinks, and Compressive Transformer (CT) using average and max pooling as the compression functions. Besides, we evaluate six tasks with average sequence lengths exceeding 8k on LongBench (Bai et al., 2024b).

Table 2: Performance and efficiency analysis on the 128k length subset of LV-Eval and InfiniteBench. The mixing/model FLOP ratio measures the relative computational cost of the token mixer or the entire model compared with the full attention baseline. For all methods except full attention, the lossless memory of attention sinks (Xiao et al., 2024c) and sliding window attention (SWA) is 32k tokens. Compressive Transformers (CT) (Rae et al., 2020) are implemented with attention sinks (Xiao et al., 2024c) and a compression function of max or average pooling.

| Base model | Token mixer | Extra param ratio | Mixing FLOP ratio | Model FLOP ratio | Memory cache ratio | LV-Eval | | | | InfiniteBench | | |
|---|---|---|---|---|---|---|---|---|---|---|---|---|
| | | | | | | cmrc -mixup | loogle-SD -mixup | dureader -mixup | Avg.* | En. QA | Zh. QA | Avg. |
| Qwen2.5-3B-Instruct | Full Attn | 0% | 100% | 100% | 100% | 7.28 | 0.89 | **13.22** | 4.41 | 7.28 | 11.75 | 9.52 |
| | Sinks + SWA | 0% | 46.6% | 59.3% | 25.6% | 7.48 | 4.59 | 11.49 | 4.59 | 8.63 | 12.31 | 10.47 |
| | CT-Max | 0% | 47.1% | 59.7% | 26.0% | 6.10 | 3.88 | 11.37 | 4.12 | 7.40 | 12.59 | 10.00 |
| | CT-Average | 0% | 47.1% | 59.7% | 26.0% | 6.95 | 4.70 | 11.40 | 4.47 | 8.30 | 13.32 | 10.81 |
| | AHN-Mamba2 | 0.4% | 46.7% | 59.4% | 26.0% | 7.84 | 5.20 | 12.35 | 5.13 | 9.29 | 15.58 | 12.44 |
| | AHN-DN | 0.4% | 46.7% | 59.4% | 26.0% | **9.41** | 5.99 | 11.49 | 5.68 | **10.61** | **16.41** | **13.51** |
| | AHN-GDN | 0.4% | 46.7% | 59.4% | 26.0% | 7.96 | **7.21** | 12.52 | **5.88** | 10.61 | 15.87 | 13.24 |
| Qwen2.5-7B-Instruct | Full Attn | 0% | 100% | 100% | 100% | 4.30 | 0.17 | 12.8 | 3.62 | 11.23 | 15.76 | 13.50 |
| | Sinks + SWA | 0% | 48.0% | 65.5% | 25.6% | 9.52 | 4.76 | 14.09 | 5.34 | 10.66 | 15.66 | 13.16 |
| | CT-Max | 0% | 48.5% | 65.8% | 26.0% | 8.35 | 4.02 | 12.34 | 4.82 | 10.56 | 15.45 | 13.00 |
| | CT-Average | 0% | 48.5% | 65.8% | 26.0% | 9.48 | 4.86 | 13.78 | 5.28 | 10.63 | 15.99 | 13.31 |
| | AHN-Mamba2 | 0.2% | 48.2% | 65.6% | 26.0% | 12.57 | 5.54 | 14.13 | 6.21 | 11.36 | 17.06 | 14.21 |
| | AHN-DN | 0.2% | 48.2% | 65.6% | 26.0% | 11.97 | **5.67** | **16.52** | **6.82** | 12.86 | 20.10 | 16.48 |
| | AHN-GDN | 0.3% | 48.2% | 65.6% | 26.0% | **12.69** | 4.71 | 15.30 | 6.54 | **13.37** | **20.48** | **16.93** |
| Qwen2.5-14B-Instruct | Full Attn | 0% | 100% | 100% | 100% | 8.79 | 1.45 | 13.84 | 4.99 | 11.23 | 13.19 | 12.21 |
| | Sinks + SWA | 0% | 49.5% | 62.3% | 25.6% | 11.96 | 7.59 | 12.23 | 5.69 | 11.62 | 13.45 | 12.54 |
| | CT-Max | 0% | 49.8% | 62.6% | 25.9% | 10.55 | 7.53 | 12.08 | 5.28 | 10.58 | 12.73 | 11.66 |
| | CT-Average | 0% | 49.8% | 62.6% | 25.9% | 11.89 | 7.41 | 12.46 | 5.64 | 10.61 | 13.28 | 11.95 |
| | AHN-Mamba2 | 0.3% | 49.7% | 62.4% | 25.9% | 14.03 | 7.20 | **15.39** | 6.43 | 14.21 | 16.20 | 15.21 |
| | AHN-DN | 0.3% | 49.7% | 62.4% | 25.9% | 13.13 | **9.14** | 14.46 | 6.50 | **16.54** | 18.42 | **17.48** |
| | AHN-GDN | 0.4% | 49.7% | 62.5% | 25.9% | **14.16** | 8.54 | 13.94 | **6.51** | 14.48 | **18.55** | 16.52 |

**Ultra-long-context.** LV-Eval is a challenging long-context benchmark, covering both single-hop QA and multi-hop QA. It introduces several design challenges, including confusing facts insertion, keyword and phrase replacement, and a keyword-recall-based metric. We evaluate all methods on the 128k-context subsets across all 11 datasets. For sliding window-based methods (SWA and AHN), we use a 32768-token lossless memory, consisting of 128-token attention sinks and a 32640-token sliding window during inference. To further validate this setting, we also test on InfiniteBench, a benchmark tailored to evaluate language models' ability to process, understand, and reason over super-long contexts. As shown in Table 2, AHN-augmented models consistently outperform SWA with attention sinks baseline across nearly all tasks. Remarkably, they also surpass the performance of full attention, demonstrating the effectiveness of the compressed memory mechanism while offering substantial computational and memory savings. We include full results in the appendix.

**Long-context.** To evaluate our models on a broader range of practical scenarios, we use Long-Bench, which features diverse tasks across multiple domains and languages, designed to rigorously test long-context understanding in more realistic scenarios. While many tasks on LongBench have relatively short inputs, we focus on six tasks with an average length exceeding 8192 tokens to create a challenging evaluation. In this setup, we constrain all methods to a fixed 8192-token lossless memory budget (128 attention sinks and an 8064-token sliding window). As reported in Table 3, AHN-augmented models again achieve consistently superior accuracy compared to both baselines. These results strongly suggest that the recurrent hidden states effectively capture and utilize historical information, leading to improved performance across diverse scenarios.

### 3.4 ABLATION STUDY

Having demonstrated the effectiveness of AHN-augmented models, we now conduct an ablation study to analyze the impact of our three design choices: the training objective, randomization of training window size, and the inference window size. For these experiments, we use AHN-GDN (Qwen2.5-7B-Instruct) as the starting point.

**Training objectives: self-distillation vs. next-token prediction.** We train AHNs using self-distillation, minimizing the KL divergence between the AHN-augmented logits and the full attention outputs. As a comparison, we also apply standard next-token prediction with cross-entropy (CE) loss, which encourages AHNs to "learn to compress" directly from data distribution. As shown in Table 4, this replacement results in a marked performance drop on LongBench. We hypothesize

Table 3: Qwen2.5-based model performance on six LongBench tasks (average sequence length > 8k). For all methods, the lossless memory of attention sinks (Xiao et al., 2024c) and sliding window attention (SWA) is 8192 tokens. Compressive Transformers (CT) (Rae et al., 2020) are implemented with attention sinks (Xiao et al., 2024c) and a compression function of max or average pooling.

| Base model | Token mixer | DuReader | HotpotQA | MuSiQue | NarrativeQA | QMSum | TriviaQA | Avg. |
|---|---|---|---|---|---|---|---|---|
| Qwen2.5-3B-Instruct | Sinks + SWA | 23.28 | 43.70 | 16.55 | 15.35 | 21.54 | 85.44 | 34.31 |
| | CT-Max | 22.81 | 40.92 | 17.22 | 16.58 | 21.07 | 85.55 | 34.03 |
| | CT-Average | 23.28 | **44.65** | 16.32 | 16.36 | 21.18 | 85.29 | 34.51 |
| | AHN-Mamba2 | 24.38 | 42.95 | 18.31 | 16.70 | 21.89 | 85.18 | 34.90 |
| | AHN-DN | 25.12 | 42.83 | **19.78** | **19.11** | **22.35** | **86.17** | **35.89** |
| | AHN-GDN | **25.47** | 42.76 | 19.31 | 18.95 | 21.85 | 84.93 | 35.55 |
| Qwen2.5-7B-Instruct | Sinks + SWA | 24.93 | 51.57 | 22.34 | 22.29 | 21.49 | 88.48 | 38.52 |
| | CT-Max | 25.08 | 50.61 | 20.65 | 23.17 | 21.34 | 88.89 | 38.29 |
| | CT-Average | 24.81 | 51.85 | 21.65 | 22.66 | 21.54 | 88.48 | 38.50 |
| | AHN-Mamba2 | 26.10 | 53.24 | 27.93 | 24.86 | **21.97** | 89.24 | 40.56 |
| | AHN-DN | 26.42 | **54.24** | **29.30** | **25.08** | 21.69 | 89.49 | **41.04** |
| | AHN-GDN | **26.97** | 54.17 | 26.83 | 24.00 | 21.80 | **89.75** | 40.59 |
| Qwen2.5-14B-Instruct | Sinks + SWA | 25.46 | 55.68 | 29.01 | 23.21 | 21.45 | 89.06 | 40.65 |
| | CT-Max | 24.63 | 54.45 | 27.78 | 22.16 | 21.16 | 88.16 | 39.72 |
| | CT-Average | 25.48 | 56.08 | 29.15 | 23.26 | 21.40 | **89.53** | 40.82 |
| | AHN-Mamba2 | 26.34 | 56.52 | 30.32 | 24.01 | 22.19 | 88.63 | 41.34 |
| | AHN-DN | **26.80** | **58.71** | **32.92** | 22.95 | 22.08 | 87.50 | 41.83 |
| | AHN-GDN | 26.51 | 58.09 | 31.40 | **24.71** | **22.35** | 88.35 | **41.90** |

Figure 4: The AHN-augmented model consistently outperforms the sliding-window attention (SWA) baseline across all window sizes on both LV-Eval and InfiniteBench (128k sequence length).

Table 4: Ablation of AHN training design choices based on Qwen2.5-7B-Instruct: (1) the training objective (2) randomized vs. fixed window.

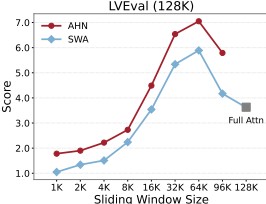
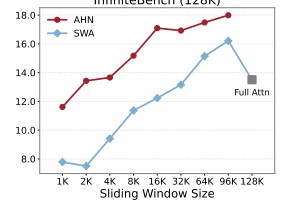

| Training target | Training window size | LongBench (Average of 6 tasks) |
|---|---|---|
| Self-distill (KL loss) | 1024 (fixed) | 38.53 |
| Next-token pred. (CE loss) | Random size | 39.59 |
| Self-distil. (KL loss) | Random size | 40.59 |

this is because CE provides sparse learning signals, and pushes the small AHN modules towards shortcuts in the training data. In contrast, self-distillation offers denser guidance over the teacher's entire output distribution, compelling AHNs to learn more generalizable context representations.

**Training window size: randomized vs. fixed.** We train AHNs using randomized sliding-window sizes to encourage a more general and robust compressive module that can adapt to varying context lengths. In contrast, models trained with a fixed window often overfit to that specific configuration and fail to generalize to unseen context lengths, leading to noticeable performance degradation, as shown in Table 4.

**Inference window size.** To further evaluate context length generalization, we fix the attention sinks to 128 tokens and test AHN-augmented models with sliding window sizes ranging from 1k to 96k on the 128k-context subsets of both LV-Eval and InfiniteBench. As shown in Figure 4, the AHN-augmented model maintains competitive performance across all window configurations, and consistently outperforms sliding window attention (SWA). Notably, as the inference window size increases from 1k to 16k, the performance improves steadily, highlighting the importance of a large attention window for extra-long context tasks. Beyond this range, however, we observe a noticeable performance drop, after 64k on LV-Eval and 96k on InfiniteBench, which may be attributed to the attention-dilution effect, where the the attention distribution becomes overly diffuse when the number of keys grows very large, weakening the model's ability to focus on relevant information. Balancing performance and computational cost, we therefore adopt a 32k sliding window as the default configuration for inference. Additional LongBench results are provided in the appendix.

## 3.5 PROBING AHN WITH GRADIENTS VISUALIZATION

Beyond benchmark performance, we seek to understand how effectively AHNs compress and exploit out-of-window information. We probe the backward dynamics of AHN-augmented models by

visualizing gradients of the self-distillation loss, which is formally defined by:

$$\frac{\partial}{\partial x_{\text{out}}} \text{KL}(f'(x_{\text{win}}, x_{\text{out}}) \parallel f(x_{\text{win}}, h_{\text{AHN}})) \tag{8}$$

where $f'(\cdot)$ and $f(\cdot)$ denote the teacher and student forward models, $h_{\text{AHN}}$ represents the compressed memory of AHN, $x_{\text{win}}$ are in-window token embeddings, and $x_{\text{out}}$ are out-of-window embeddings. Out-of-window tokens with small gradient magnitudes indicate that their information has already been well captured in AHN's compressed memory.

Figure 5 shows an example from AceMath-Instruct-Training-Data (Liu et al., 2025). The full sequence has 811 tokens, and we evaluate the AHN-augmented model using a 512-token sliding window (AHN activates once the context surpasses 512 tokens). The snippet shows the gradients for the first 139 tokens. As illustrated in Figure 5, AHN tends to preserve the information of mathematical symbols and numbers while neglecting less critical ones such as pronouns and special tokens, demonstrating its AHNs can learn to prioritize more informative tokens for storage.

```
<|im_start|>system You are a h
elpful assistant.<|im_end|>

<|im_start|>user Given the acu
te angles \( \) \( B \) such \
( \ (A + B) 2 \tan A \), what
is the maximum value of \( \ta
n B \)?<|im_end|>

<|im_start|>assistant Given th
e equation \(\tan (A + B) 2 \t
an A), we start by using the t
angent addition formula: \[ \t
an (A + B) \frac{\tan A + \tan
 B}{ \tan A \tan B} \] Substit
uting the given equation we ge
t: \[ \frac{\tan A + \tan B}{1
 - \tan \ B} = 2 \tan A \]
```

Figure 5: Green regions mark tokens with low L2 gradient magnitudes, indicating they are preferentially selected by AHN to store in the compressed memory; red denotes the opposite.

## 4 RELATED WORK

### 4.1 MEMORY IN NEURAL NETWORKS

Memory mechanisms play a crucial role in enabling neural networks to process and retain information over time, which is essential for tasks that require understanding of temporal dependencies, sequential data, or context preservation. Traditional feedforward neural networks lack the capability to maintain information across time steps, which limits their effectiveness in tasks such as language modeling, sequence prediction, and reasoning. To address this limitation, Recurrent Neural Networks (RNNs) are introduced (Werbos, 1988; Jordan, 1986; Elman, 1990; Hopfield, 1982; 1984). RNNs maintain a hidden state that is updated at each time step, allowing information to persist across sequences. However, vanilla RNNs suffer from issues such as vanishing and exploding gradients, making it difficult to capture long-term dependencies (Bengio et al., 1994). To mitigate these problems, more advanced architectures like Long Short-Term Memory (LSTM) networks (Hochreiter & Schmidhuber, 1997) and Gated Recurrent Unit (GRU) (Cho et al., 2014) are proposed. These models incorporate gating mechanisms that regulate the flow of information, enabling them to learn longer-term dependencies more effectively. Because these RNN-like models maintain a fixed-size memory and a consistent memory update cost for each input token, they are highly efficient for processing long sequences. Therefore, our AHNs are designed within the RNN paradigm to inherit this advantageous property.

Beyond RNN-based architectures, memory-augmented neural networks have been developed to further enhance the memory capacity of neural models. For example, the Neural Turing Machine (NTM) (Graves et al., 2014) and the Differentiable Neural Computer (DNC) (Graves et al., 2016) introduce external memory modules that the network can read from and write to, allowing for more complex reasoning and algorithmic tasks. Over the past decade, attention mechanisms (Bahdanau et al., 2015) have revolutionized the way neural networks handle memory. The Transformer architecture (Vaswani et al., 2017), which relies entirely on self-attention mechanisms, enables direct access to all previous states in a sequence, providing a form of memory that is both lossless and scalable. This has led to significant improvements in various domains (Radford et al., 2018; 2019; Devlin et al., 2019; Dosovitskiy et al., 2021), and has spurred the emergence of new technological paradigms and innovations (OpenAI, 2023; 2024a;b; Guo et al., 2025), such as In-Context Learning (Brown et al., 2020) and Chain-of-Thought (CoT) reasoning (Wei et al., 2022). However, modeling long sequences exacerbates the quadratic computational complexity cost of attention mechanisms (Child et al., 2019). Our proposed AHNs address this challenge by employing an RNN-like network to compress the historical KV cache.

## 4.2 MEMORY MANAGEMENT

RNN-like models (Elman, 1990; Hochreiter & Schmidhuber, 1997; Cho et al., 2014; Katharopoulos et al., 2020; Peng et al., 2023; Sun et al., 2023; Gu & Dao, 2024; Yang et al., 2024c; Zhang et al., 2024b; Yang et al., 2024d; Dao & Gu, 2024; Beck et al., 2024; Yang et al., 2025) maintain memory through a fixed-size hidden state, regardless of input sequence length. Therefore, memory caching is not a major concern for these architectures. In contrast, Transformers store key-value (KV) pairs for every token in the input sequence, resulting in linear growth of the KV cache with sequence length. This results in significant memory consumption and presents a major challenge for processing long sequences. To mitigate this issue, various approaches have been proposed (Li et al., 2024a), including KV cache selection (Ge et al., 2024; Li et al., 2024b; Xiao et al., 2024c; Han et al., 2024; Zhang et al., 2023; Liu et al., 2023; Adnan et al., 2024; Xiao et al., 2024a; Tang et al., 2024), budget allocation (Cai et al., 2024; Yang et al., 2024b; Feng et al., 2024; Xiao et al., 2024b), merging (Nawrot et al., 2024; Wan et al., 2024; Wang et al., 2024b; Liu et al., 2024b), quantization (Yao et al., 2022; Sheng et al., 2023; Hooper et al., 2024; Xiao et al., 2023; Lin et al., 2024; Shao et al., 2024), low-rank decomposition (Yu et al., 2024; Dong et al., 2024), external memory (Packer et al., 2023; Wang et al., 2025), and neural architecture design (Shazeer, 2019; Ainslie et al., 2023; Liu et al., 2024a; Hua et al., 2022; Sun et al., 2024; Yen, 2024; Wu et al., 2022; Munkhdalai et al., 2024). Among them, a straightforward strategy is to use a sliding window for attention (Vaswani et al., 2017), but this method discards KV pairs outside the window, thereby losing long-range context. Sparse Transformers (Child et al., 2019) address this by retaining KV pairs at specific pattern positions to capture long-range dependencies, but still drop portions of the KV cache, potentially missing important information. Transformer-XL (Dai et al., 2019) introduces a segment-level recurrence mechanism by caching the last segment of hidden states as a First-In, First-Out (FIFO) memory. Compressive Transformer (Rae et al., 2020) extends this by compressing older memories into a secondary FIFO memory, but it still discards memory once the slots are full. In contrast, AHNs adopt an RNN-like paradigm that continually compresses KV pairs outside the sliding window into a lifelong compressed memory, rather than discarding them outright (Lieber et al., 2024; Munkhdalai et al., 2024; Ren et al., 2025). AHNs (like AHN-GDN (Yang et al., 2025)) can also dynamically control memory decay (Dao & Gu, 2024; Schlag et al., 2021; Yang et al., 2024d; 2025). Recent studies integrate RNNs and attention either in interleaved layers (Lieber et al., 2024; Ren et al., 2025; Dao & Gu, 2024; Yang et al., 2025; Li et al., 2025a) or within a single layer (Munkhdalai et al., 2024; Behrouz et al., 2024; Li et al., 2025b). By contrast, we abstract the compression module as an AHN concept, yielding a more general memory framework. We employ a sliding-window attention mechanism, activating AHNs whenever a token leaves the window. Additionally, we introduce a simple self-distillation scheme that trains AHNs efficiently.

Compared with recent attention-RNN-Hybrid works (Munkhdalai et al., 2024; Wang et al., 2024a; Zhang et al., 2025) and concurrent work (Irie et al., 2025), the goal of the AHN memory framework is to leverage the efficiency of RNNs specifically to address the computational bottleneck of attention on extra-long sequences. The distinct insight introduced by AHNs is to employ a large sliding-window size (*e.g.*, 32k) for attention, such that the RNN-like AHN modules only activate when the sequence length exceeds this window. This design provides two major advantages: 1) It highlights the efficiency benefits of RNNs on extra-long context tasks (e.g., 128k), achieving substantial FLOP and memory-cache savings. In contrast, for short-context tasks where attention is already efficient, introducing RNNs provides no efficiency gain. 2) It requires no additional effort to preserve attention performance on short-context tasks, because AHNs remain inactive in these regimes, and the model behaves exactly as a pure attention-based Transformer.

## 5 CONCLUSION

We introduce Artificial Hippocampus Networks (AHNs), a novel class of lightweight architectural components that enhance Transformer models for efficient long-sequence processing. AHNs address the efficiency limitation of standard transformers by maintaining a sliding window of KV cache as lossless memory while transforming out-of-window information into a fixed-size compressed memory. This approach enables AHN-augmented models to achieve constant memory and computational complexity per token over long sequences. Experiments demonstrate that AHNs can significantly reduce both memory cache size and computation while maintaining competitive performance on long-context benchmarks.

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

# A  APPENDIX

## A.1  AHN INSTANTIATION

This section describes how to instantiate AHNs with Mamba2 (Dao & Gu, 2024) and DateNet (DN) (Schlag et al., 2021; Yang et al., 2024d). For the AHN-Mamba2 instance, the compressed memory update rule is

$$
\begin{aligned}
h_{t-W} &= \text{AHN-Mamba2}((k_{t-W}, v_{t-W}), h_{t-W-1}, x_{t-W}) \\
&= \exp(-\Delta(x_{t-W})A)h_{t-W-1} + \Delta(x_{t-W-1})k_{t-W}^T v_{t-W}
\end{aligned}
\tag{9}
$$

As for AHN-DN, the update rule can be expressed as

$$
\begin{aligned}
h_{t-W} &= \text{AHN-DN}((k_{t-W}, v_{t-W}), h_{t-W-1}, x_t) \\
&(\mathbf{I} - \beta(x_{t-W})k_{t-W}^T k_{t-W})h_{t-W-1} + \beta(x_{t-W})k_{t-W}^T v_{t-W}
\end{aligned}
\tag{10}
$$

The output rule of AHN-Mamba2 and AHN-DN are the same as AHN-GDN, as shown in Equation 6.

We also provide an illustration of AHN-augmented networks with attention sinks (Xiao et al., 2024c), as shown in Figure 6.

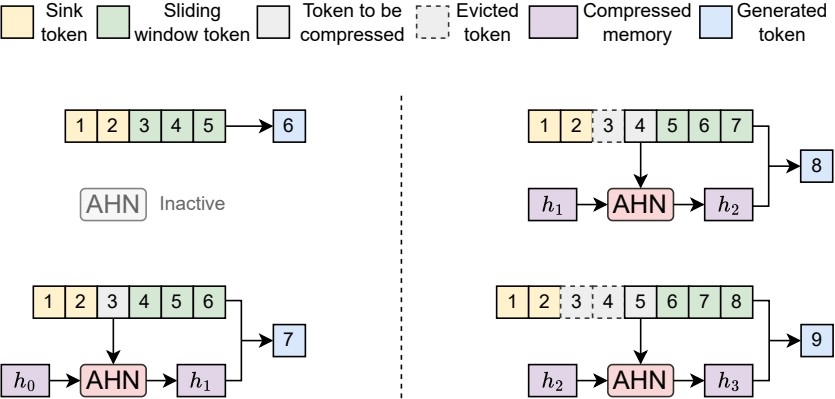

Figure 6: Illustration of the model augmented with Artificial Hippocampus Networks (AHNs). In this example, the number of attention sinks is 2, and the sliding window length is 3. When the input sequence length is less than or equal to the sum of attention sinks and the window length, the model operates identically to a standard Transformer. For longer sequences, AHNs continually compress the token outside the window into a compact memory representation. The model then utilizes the lossless information within the attention sinks and the sliding window, as well as the compressed memory to generate the next token.

## A.2  ADDITIONAL BENCHMARK RESULTS

This section further examines the effectiveness of AHNs in long-context scenarios, presenting additional benchmark results, while also acknowledging their inherent limitations on exact-recall tasks due to the lossy nature of compressed memory.

**LV-Eval** (Yuan et al., 2024). We present complete results on all 11 LV-Eval tasks under the 128k context setting. All models are configured with 32768 tokens of lossless memory, including 128-token attention sinks and a 32640-token sliding window.

**RULER** (Hsieh et al., 2024) is a comprehensive benchmark that extends the standard needle-in-a-haystack (NIAH) (Kamradt, 2023) paradigm by introducing increased task difficulty and additional categories. We evaluate an AHN-augmented model (AHN-GDN) on all NIAH tasks within the RULER-128k subset, using Qwen2.5-7B-Instruct as the base model. For a fair comparison, both AHN-GDN and sliding window attention with attention sinks are configured with 128 attention

sinks and a 32640-token sliding window. As shown in Table 5, AHN-GDN performs on par with sliding window attention but markedly worse than full attention on exact-recall tasks. This reflects the inherent trade-off of lossy compression: while AHN-augmented models enable efficient long-context reasoning, they inevitably struggle on tasks that require exact-recall from the compressed memory. This limitation suggests opportunities for future research, such as memory management that preserves critical information in lossless memory while leveraging compression for efficiency.

Table 5: Performance on advanced needle-in-a-haystack (NIAH) tasks performance from RULER-128k. Both sliding window approaches use 128 attention sinks with a 32640 sliding window.

| Method | single_1 | single_2 | single_3 | multikey_1 | multikey_2 | multikey_3 | multivalue | multiquery |
|---|---|---|---|---|---|---|---|---|
| Full Attn | 98.60 | 97.20 | 98.40 | 89.20 | 23.60 | 23.20 | 55.40 | 85.45 |
| Sinks + SWA | 26.80 | 25.40 | 28.00 | 27.80 | 10.60 | 9.00 | 22.95 | 24.00 |
| AHN-GDN | 26.80 | 25.20 | 28.20 | 27.40 | 11.40 | 8.60 | 23.45 | 23.35 |

Table 6: Complete results on all 21 tasks in the 128k subset of LV-Eval. All sliding window-based methods use a lossless memory of 32768 tokens, consisting of 128 attention sinks and a 32640-token sliding window.

| Model | Dataset | Full Attn | Sinks + SWA | CT-Max | CT-Average | AHN-Mamba2 | AHN-DN | AHN-GDN |
|---|---|---|---|---|---|---|---|---|
| | **Average** | 4.41 | 4.59 | 4.12 | 4.47 | 5.13 | _5.68_ | **5.88** |
| Qwen2.5-3B-Instruct | cmrc_mixup | 7.28 | 7.48 | 6.10 | 6.95 | 7.84 | **9.41** | _7.96_ |
| | dureader_mixup | **13.22** | 11.49 | 11.37 | 11.4 | 12.35 | 11.71 | _12.52_ |
| | factrecall_en | 6.88 | 3.34 | 3.86 | 3.59 | 5.58 | _9.22_ | **12.51** |
| | factrecall_zh | _2.80_ | 1.28 | 1.37 | 1.18 | 1.57 | **4.19** | 1.79 |
| | hotpotwikiqa_mixup | 0.09 | 0.30 | 0.08 | 0.48 | **1.11** | 0.06 | _0.65_ |
| | lic_mixup | 7.68 | 6.86 | 6.39 | 6.49 | **8.13** | _7.78_ | 7.38 |
| | loogle_CR_mixup | 0.06 | _2.24_ | 1.61 | **2.28** | 1.55 | 1.65 | 1.96 |
| | loogle_MIR_mixup | 0.00 | 0.64 | 0.47 | 0.58 | **1.39** | _1.14_ | 1.06 |
| | loogle_SD_mixup | 0.89 | 4.59 | 3.88 | 4.70 | 5.20 | _5.99_ | **7.21** |
| | multifieldqa_en_mixup | 0.00 | _0.33_ | **0.43** | 0.08 | 0.00 | 0.00 | 0.19 |
| | multifieldqa_zh_mixup | 9.59 | **11.91** | 9.74 | 11.41 | _11.72_ | 11.31 | 11.42 |
| | **Average** | 3.62 | 5.34 | 4.82 | 5.28 | 6.21 | **6.83** | _6.54_ |
| Qwen2.5-7B-Instruct | cmrc_mixup | 4.30 | 9.52 | 8.35 | 9.48 | _12.57_ | 11.97 | **12.69** |
| | dureader_mixup | 12.80 | 14.09 | 12.34 | 13.78 | 14.13 | **16.52** | _15.30_ |
| | factrecall_en | 5.33 | 4.65 | 4.67 | 4.65 | **5.84** | _5.74_ | 5.14 |
| | factrecall_zh | 0.80 | 1.29 | 1.11 | 1.35 | 1.43 | **2.05** | _1.68_ |
| | hotpotwikiqa_mixup | 0.24 | 0.69 | 0.48 | _0.82_ | 0.16 | **0.99** | 0.76 |
| | lic_mixup | 3.40 | _10.19_ | 8.49 | 10.07 | 9.27 | 8.73 | **10.63** |
| | loogle_CR_mixup | 0.57 | 0.50 | 0.81 | 0.47 | _2.26_ | **2.59** | 1.58 |
| | loogle_MIR_mixup | 0.00 | 0.71 | 1.08 | 0.92 | 0.91 | **3.08** | _2.70_ |
| | loogle_SD_mixup | 0.17 | 4.76 | 4.02 | 4.86 | _5.54_ | **5.67** | 4.71 |
| | multifieldqa_en_mixup | 0.00 | _0.47_ | **0.71** | 0.45 | 0.00 | 0.28 | 0.06 |
| | multifieldqa_zh_mixup | 12.24 | 11.90 | 10.93 | 11.27 | 16.18 | **17.49** | _16.74_ |
| | **Average** | 4.99 | 5.69 | 5.28 | 5.64 | 6.43 | _6.50_ | **6.51** |
| Qwen2.5-14B-Instruct | cmrc_mixup | 8.79 | 11.96 | 10.55 | 11.89 | _14.03_ | 13.13 | **14.16** |
| | dureader_mixup | 13.84 | 12.23 | 12.08 | 12.46 | **15.39** | _14.46_ | 13.94 |
| | factrecall_en | **4.31** | 0.45 | 0.77 | 0.45 | _1.19_ | 0.30 | 0.15 |
| | factrecall_zh | **0.22** | 0.07 | 0.13 | 0.00 | _0.15_ | 0.00 | 0.00 |
| | hotpotwikiqa_mixup | 0.00 | _0.64_ | 0.53 | _0.64_ | 0.33 | **0.67** | 0.49 |
| | lic_mixup | _11.96_ | 10.18 | 9.52 | 10.19 | 11.57 | **12.17** | 11.13 |
| | loogle_CR_mixup | 0.3 | **3.64** | 2.74 | 3.57 | 3.60 | 2.34 | **3.64** |
| | loogle_MIR_mixup | 0.94 | _1.56_ | 1.38 | 1.36 | **1.65** | 1.19 | 0.65 |
| | loogle_SD_mixup | 1.45 | 7.59 | 7.53 | 7.41 | 7.20 | **9.14** | _8.54_ |
| | multifieldqa_en_mixup | 0.00 | 0.41 | 0.39 | 0.06 | 0.60 | **1.08** | _0.94_ |
| | multifieldqa_zh_mixup | 13.10 | 13.82 | 12.50 | 14.05 | 14.97 | _17.06_ | **17.94** |

## A.3 ADDITIONAL SLIDING WINDOW SIZE GENERALIZATION ON LONGBENCH

The sequence lengths of LongBench tasks are substantially shorter than those in LV-Eval and InfiniteBench. We therefore select six relatively long tasks from LongBench, whose average sequence lengths range from 8k to 18k. To evaluate the context-length generalization ability of AHN-augmented models on these tasks, we fix the attention-sink size to 128 tokens and vary the sliding-window size from 896 to 8064. We compare AHN-augmented models against both Sliding Window

Figure 7: AHN modules demonstrate strong context generalization capacity on LongBench.

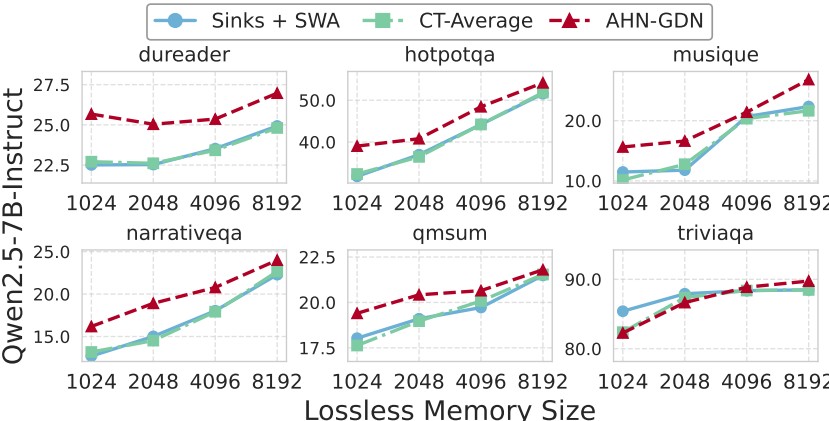

Table 7: One-step training FLOPs ($10^{17}$) under the setting of AdamW optimizer, next-token prediction, full-parameter tuning, batch size 128, sequence length 24k, and sliding-window size 8k.

| Model | 3B | 7B | 14B |
|---|---|---|---|
| Full attention | 0.6348 | 1.1519 | 2.5396 |
| Attention Sinks + SWA | 0.5405 | 1.0334 | 2.2252 |
| AHN-GDN | 0.5422 | 1.0359 | 2.2319 |

Attention (SWA) and Compressive Transformers using average pooling (CT-Average). As shown in Figure 7, AHN-augmented models consistently outperform these baselines across different inference window sizes.

### A.4 DETAILED EFFICIENCY NUMBERS

Due to the space limit, we only show the relative ratio of FLOPs and memory cache in Table 2. The detailed numbers are shown in the Table 9.

For trianing, our method uses a self-distillation training strategy in which only the AHN parameters are optimized, while all parameters of the base LLM remain frozen. When training on the ChatQA 2 dataset (Xu et al., 2025) with 1B tokens, it takes only 10 hours on 32 A100 GPUs to train AHNs for the Qwen2.5-7B model. Although the maximum training sequence length is 24k, the resulting model generalizes to much longer sequences (e.g., 128k) during inference. Importantly, our method does not require re-training the base LLM. To compare the training efficiency of sliding-window attention, AHN-augmented models, and full attention, we calculate their training FLOPs under a unified setting: AdamW optimizer, next-token prediction, full-model training, batch size 128, 24k sequence length, and 8k sliding window size. The FLOPs per training step are summarized in Table 7.

### A.5 PURE RNN BASELINE

The goal with AHNs is not to replace attention with RNNs, but to leverage the efficiency of RNNs to address the quadratic-complexity bottleneck of attention on extra-long sequences. The AHN framework fundamentally relies on a large sliding-window attention (SWA) to preserve the strengths of attention on short and medium sequences. Removing SWA and using only RNNs would break the design principle of our memory framework and result in a model that cannot function as intended. We conduct an ablation experiment by removing SWA entirely and keeping only the AHN (RNN) module, and the results are shown in 8. These results confirm that pure RNNs alone are insufficient in our memory framework for long-context tasks.

Table 8: Module ablation for the AHN memory framework on Qwen2.5-7B (AHN-GDN variant).

| Module | LV-Eval Avg | InfiniteBench Avg |
|---|---|---|
| Sinks + SWA | 5.34 | 13.16 |
| Pure RNN | 0.04 | 1.19 |
| AHN framework (Sinks + SWA + RNN) | 6.54 | 16.93 |

Table 9: Inference efficiency numbers for 128k sequence length. The mixing/model FLOP ratio measures the relative computational cost of the token mixer or the entire model compared with the full attention baseline. For all methods except full attention, the lossless memory of attention sinks (Xiao et al., 2024c) and sliding window attention (SWA) is 32k tokens. Compressive Transformers (CT) (Rae et al., 2020) are implemented with attention sinks (Xiao et al., 2024c) and a compression function of max or average pooling.

| Base model | Token mixer | Extra param (M) | Extra param ratio | Mixing FLOPs ($10^{15}$) | Mixing FLOP ratio | Model FLOPs ($10^{15}$) | Model FLOP ratio | Memory cache (GB) | Memory cache ratio |
|---|---|---|---|---|---|---|---|---|---|
| Qwen2.5-3B-Instruct | Full Attn | 0 | 0% | 2.50 | 100% | 3.29 | 100% | 9.44 | 100% |
| | Sinks + SWA | 0 | 0% | 1.17 | 46.6% | 1.95 | 59.3% | 2.42 | 25.6% |
| | CT-Max | 0 | 0% | 1.18 | 47.1% | 1.96 | 59.7% | 2.45 | 26.0% |
| | CT-Average | 0 | 0% | 1.18 | 47.1% | 1.96 | 59.7% | 2.45 | 26.0% |
| | AHN-Mamba2 | 11.9 | 0.4% | 1.17 | 46.7% | 1.95 | 59.4% | 2.45 | 26.0% |
| | AHN-DN | 11.8 | 0.4% | 1.17 | 46.7% | 1.95 | 59.4% | 2.45 | 26.0% |
| | AHN-GDN | 13.0 | 0.4% | 1.17 | 46.7% | 1.95 | 59.4% | 2.45 | 26.0% |
| Qwen2.5-7B-Instruct | Full Attn | 0 | 0% | 3.23 | 100% | 4.87 | 100% | 14.7 | 100% |
| | Sinks + SWA | 0 | 0% | 1.55 | 48.0% | 3.19 | 65.5% | 3.76 | 25.6% |
| | CT-Max | 0 | 0% | 1.57 | 48.5% | 3.20 | 65.8% | 3.81 | 26.0% |
| | CT-Average | 0 | 0% | 1.57 | 48.5% | 3.20 | 65.8% | 3.81 | 26.0% |
| | AHN-Mamba2 | 18.6 | 0.2% | 1.56 | 48.2% | 3.19 | 65.6% | 3.81 | 26.0% |
| | AHN-DN | 18.5 | 0.2% | 1.56 | 48.2% | 3.19 | 65.6% | 3.81 | 26.0% |
| | AHN-GDN | 21.3 | 0.3% | 1.56 | 48.2% | 3.19 | 65.6% | 3.81 | 26.0% |
| Qwen2.5-14B-Instruct | Full Attn | 0 | 0% | 8.83 | 100% | 11.83 | 100% | 50.33 | 100% |
| | Sinks + SWA | 0 | 0% | 4.37 | 49.5% | 7.38 | 62.3% | 12.88 | 25.6% |
| | CT-Max | 0 | 0% | 4.40 | 49.8% | 7.41 | 62.6% | 13.01 | 25.9% |
| | CT-Average | 0 | 0% | 4.40 | 49.8% | 7.41 | 62.6% | 13.01 | 25.9% |
| | AHN-Mamba2 | 51.4 | 0.3% | 4.38 | 49.7% | 7.39 | 62.4% | 13.01 | 25.9% |
| | AHN-DN | 51.1 | 0.3% | 4.38 | 49.7% | 7.39 | 62.4% | 13.01 | 25.9% |
| | AHN-GDN | 61.0 | 0.4% | 4.38 | 49.7% | 7.39 | 62.5% | 13.01 | 25.9% |

### A.6 COMPARISON TO RECENT AND CONCURRENT ATTENTION-RNN-HYBRID WORKS

Besides the discussions in Section 4, here are the detailed differences between AHNs and recent attention-RNN-hybrid works, Infini-attention (Munkhdalai et al., 2024), MiL (The Mamba in the Llama) (Wang et al., 2024a), LoLCATs (Zhang et al., 2025) and concurrent work HQLT (Irie et al., 2025):

**Model architecture.** Infinite-attention (Munkhdalai et al., 2024) performs chunk-wise attention and updates its recurrent memory in a chunk-wise way. In contrast, AHN is built on a standard decoder-only autoregressive Transformer and updates its compressed memory in a token-wise manner. This token-wise design allows AHN to be integrated seamlessly into existing popular base models, and it enables flexible configuration of the sliding-window size for attention according to available hardware memory. Different from MiL (Wang et al., 2024a), which distills all attention layers into a linear RNN, AHN only activates when the sequence length exceeds a large sliding window. Different from the small attention window of 64 used in LoLCATs (Zhang et al., 2025) and HQLT (Irie et al., 2025), and 2048 used in Infini-attention (Munkhdalai et al., 2024), AHN adopts a much larger 32k sliding-window size during inference. Since quadratic attention remains efficient for short and medium sequences, the quadratic-complexity bottleneck only appears when sequences become extra long. This motivates us to set a substantially larger attention window so that AHNs activate only when the sequence length exceeds this window and attention begins to encounter efficiency issues.

**Target tasks and evaluation setting.** MiL (Wang et al., 2024a), LoLCATs Zhang et al. (2025), and HQLT (Irie et al., 2025) evaluate models mainly on short-context tasks (e.g., ARC and HellaSwag with sequence length $< 128$), where linear RNNs cannot demonstrate efficiency advantages. Besides lacking efficiency gains, these methods must also make additional efforts to match the performance of attention on these short-context tasks. In contrast, AHN targets extra-long-context tasks (e.g., LV-Eval and InfiniteBench with 128k sequence length). AHN does not activate when the sequence length is shorter than the 32k window size, so the model operates exactly as a standard Transformer. In other words, the performance of our method on short-context tasks is identical to full attention, and no extra effort is required to match attention's performance. For extra-long-context tasks such as LV-Eval and InfiniteBench, AHN not only achieves performance comparable to full attention, but also significantly reduces FLOPs and memory cache.

**Training method.** Infini-attention (Munkhdalai et al., 2024) does not disclose its overall training cost; it only reports training for 30K steps with a batch size of 64 before fine-tuning on the passkey retrieval task. MiL (Wang et al., 2024a) needs to train the whole token mixer parameters with 20B tokens, and HQLT (Irie et al., 2025) trains the entire model from scratch with 15B tokens. In contrast, we freeze the base model's all parameters and only train the newly added AHN parameters (about 0.4% of the base model) using only 1B tokens, with only 740 update steps with batch size of 128. Compared to LoLCATs (Zhang et al., 2025), which trains models through multiple stages of Attention Transfer and Low-rank Linearizing, AHN uses a simple one-stage self-distillation process.

### A.7 LIMITATIONS AND FUTURE WORKS

While AHNs strike an effective balance between computational efficiency and memory fidelity, their fixed-size compressed memory inevitably entails some information loss and may impair performance on tasks that require exact recall, as detailed in the appendix. Furthermore, since our study adopts a parameter-efficient self-distillation setup, performance remains capped by the underlying base models' capacity. Future work may explore stronger recall mechanisms and full-parameter training to further unlock the potential of AHNs. For application scenarios, the AHN framework opens up opportunities in long-context domains with sparse information or constrained resources, such as lifelong learning, streaming video processing, and deployment on edge devices.

