# OpenReview forum: "Artificial Hippocampus Networks for Efficient Long-Context Modeling"
_ICLR.cc/2026/Conference — Submitted to ICLR 2026_

### Official Review · Reviewer_YqVp · 2025-10-21

**Soundness:** 3
**Presentation:** 3
**Contribution:** 1
**Rating:** 2
**Confidence:** 4

**Summary:**

This work presents Artificial Hippocampus Networks, replacing the attention mechanism with a combination of sliding window attention and a linear RNN (Mamba2 or (Gated) Delta-Net), via distillation from a pretrained model. Experiments show that capabilities can be approximately recovered compared to the full attention original, with strong speed ups and memory reduction (constant memory use), especially on long context benchmarks.

**Strengths:**

The work puts a nice focus on long-context capabilities as a typical weak point of linear RNNs.
It even out-performs Full Attention base model on Long Context Tasks in some cases.
The proposed randomized sliding window size results in a slight performance improvement.

**Weaknesses:**

The work does not compare to very similar existing methods (LoLCaTs) [1], resulting in a lack of novelty.
The work is missing a comparison to a pure RNN, removing sliding window attention overall. If this performed similarly, also the neuro-biological motivation should be questioned.

[1] Zhang et al. (2024): https://arxiv.org/pdf/2410.10254

[2] Wang et al. (2024): https://arxiv.org/pdf/2408.15237

**Questions:**

Could you cite and explain the difference to [1] / [2] as a very similar related work?
If the sliding window attention was removed completely, how would this simpler replacement perform?

---

> ### Author Response · Authors · 2025-11-24
>
> We sincerely appreciate the reviewer’s time and feedback. We have updated the paper with further clarifications and discussion (highlighted in blue). Below, we address each question individually:
>
> **Q1. Comparison to MiL (The Mamba in the Llama: https://arxiv.org/abs/2408.15237) and LoLCATs (https://arxiv.org/abs/2410.10254).**
>
> A: Thank you for your comments. Below we summarize the main differences between AHNs and MiL and LoLCATs:
>
> **Model architecture**
>
> Different from MiL, which distills the entire attention layer into a linear RNN, AHN only activates when the sequence length exceeds a large sliding window. Different from the small attention window of 64 used in LoLCATs, AHN adopts a much larger 32k sliding-window size during inference. Since quadratic attention remains efficient for short and medium sequences, the quadratic-complexity bottleneck only appears when sequences become extra long. This motivates us to set a substantially larger attention window so that AHNs activate only when the sequence length exceeds this window and attention begins to encounter efficiency issues.
>
> **Target Tasks and Evaluation Setting**
>
> MiL and LoLCATs evaluate models mainly on short-context tasks (e.g., ARC and HellaSwag with sequence length <128), where linear RNNs cannot demonstrate efficiency advantages. Besides lacking efficiency gains, these methods must also make additional efforts to match the performance of attention on these short-context tasks.
>
> In contrast, AHN targets extra-long-context tasks (e.g., LV-Eval and InfiniteBench with 128k sequence length). AHN does not activate when the sequence length is shorter than the 32k window size, so the model operates exactly as a standard Transformer. In other words, the performance of our method on short-context tasks is identical to full attention, and no extra effort is required to match attention’s performance. For extra-long-context tasks such as LV-Eval and InfiniteBench, AHN not only achieves performance comparable to full attention, but also significantly reduces FLOPs and memory cache.
>
> **Training method**
>
> Compared with MiL, AHN’s training method is more efficient. MiL needs to train the token mixer parameters with 20B tokens. In contrast, AHN freezes the basemodel’s all parameters and only trains the newly added AHN parameters (about 0.4% relative to the base model) with only 1B tokens. Compared to LoLCATs, which trains models through multiple stages of Attention Transfer and Low-rank Linearizing, AHN uses a simple one-stage self-distillation process.
>
> **In summary**, compared with these works, the goal of the AHN memory framework is to leverage the efficiency of RNNs specifically to address the computational bottleneck of attention on extra-long sequences. The distinct insight introduced by AHNs is to employ a large sliding-window size (eg, 32k) for attention, such that the RNN-like AHN modules only activate when the sequence length exceeds this window. This design provides two major advantages:
>
> 1) It highlights the efficiency benefits of RNNs on extra-long context tasks (e.g., 128k), achieving substantial FLOP and memory-cache savings. In contrast, for short-context tasks where attention is already efficient, introducing RNNs provides no efficiency gain.
>
> 2) It requires no additional effort to preserve attention performance on short-context tasks, because AHNs remain inactive in these regimes, and the model behaves exactly as a pure attention-based Transformer.
>
> Thanks for your suggestion. We have included these discussions in the revised paper.
>
> **Q2: Missing a comparison to a pure RNN, removing sliding window attention overall**
>
> A: Thank you for your comments. The goal with AHNs is not to replace attention with RNNs, but to leverage the efficiency of RNNs to address the quadratic-complexity bottleneck of attention on extra-long sequences. The AHN framework fundamentally relies on a large sliding-window attention (SWA) to preserve the strengths of attention on short and medium sequences. Removing SWA and using only RNNs would break the design principle of our memory framework and result in a model that cannot function as intended. Following the reviewer’s suggestion, we conduct an ablation experiment by removing SWA entirely and keeping only the AHN (RNN) module. The results are shown below:
>
> Table: Module ablation for the AHN memory framework on Qwen2.5-7B (AHN-GDN variant).
>
> | Module                            | LV-Eval Avg | InfiniteBench Avg |
> |-----------------------------------|-------------|-------------------|
> | Sinks + SWA                       | 5.34        | 13.16             |
> | Pure RNN                          | 0.04        | 1.19              |
> | AHN framework (Sinks + SWA + RNN) | 6.54        | 16.93             |
>
> These results confirm that pure RNNs alone are insufficient in our memory framework for long-context tasks. Thanks for your suggestion. We have included these discussions and results in the revised paper.

---

### Official Review · Reviewer_RgMK · 2025-10-30

**Soundness:** 2
**Presentation:** 2
**Contribution:** 2
**Rating:** 2
**Confidence:** 4

**Summary:**

This paper introduces Artificial Hippocampus Networks (AHN), a method that can be applied to pretrained models in order to enhance their long-context performance while reducing their memory requirements.
AHN consist of a lossless memory in form of a sliding window attention layer and a compressed memory in form of a (linear) RNN, like for e.g. Mamba-2 or Gated Delta Net.
The authors propose to augment pretrained models with AHN by training the models with a KL-Divergence objective.
In their experiments AHN augmented Transformer models show improvements on numerous long context tasks.

**Strengths:**

- AHN augmented Transformer models provide efficiency gains
- The ablation on different training objectives is important to support the choice of solely train with the KL-Loss
- The illustrative example is a very good motivation and a good comparison to the unchanged base model

**Weaknesses:**

- Even though the relation to the human brain is a nice motivation, at the end of the introduction the reader is left with the question: what is the difference to a hybrid model between a RNN and attention?
I suggest that the authors add information about the key descriptive components of AHN and describe their method in more detail already in Fig1/Abstract/Intro? (e.g. Hybrid SWA + Linear attention, KL loss / distillation, application to pretrained models). As it is now it just seems too broad.

- L. 401, 412: The author chose a very large lossless sliding window. Since the results in Table 2 suggest that the AHN variants outperform full attention, it would be interesting to investigate the impact of the sliding window size on these benchmarks, too (not only on LongBench as in Fig.4)

- The authors could provide a more comprehensive eval comparing the pretrained basemodel with the AHN augmented variants on other standard (not necessarily long context) pretraining tasks, such as MMLU(-PRO) or GSM8k or GPQA.

- It seems that there are some important implementation details missing: From the “extra param ratio” in Table 2 one could assume that the extra parameters are learnable gates etc. not present in the pretrained transformer. Is this true? I could not find this information prominently written in the paper. So my questions remain:
  - So the QKV projections for Sliding window attention and Mamba2/GDN are shared?
  - Could the authors specify what are the “unfrozen AHN parameters” (Fig. 2)

- AHN seem not novel: Distillation with KL divergence of pretrained models has been explored, hybrid models between sliding window and linear attention also have been explored before (See for e.g. https://arxiv.org/abs/2506.00744, https://arxiv.org/abs/2410.10254 , https://arxiv.org/abs/2408.15237).
  - Could the authors emphasize the new insights stemming from the suggested AHNs?

In my view the aforementioned weaknesses outweigh the strengths. Therefore I am inclined to recommend rejection of the paper.

**Questions:**

- Figure 1 is placed before the abstract. I recommend double checking whether this is in line with the style guides for ICLR.
- I would suggest to move the related work to the end, since the introduction already covers a lot of related work. It seems a bit repetitive in this order.
- L. 353: What sliding window sizes were used? What is meant by randomizing the starting position?
- Section 4.5: From which part in the sequence is the snippet taken? Even though the motivation to try to understand how AHNs compress information, the empirical data provided by Figure 5 is not convincing and at least needs further clarifications (see question above).

- Finally, I suggest incorporating a comparison to this work ( https://arxiv.org/abs/2506.00744 ) which seems very similar.

---

> ### Author Response · Authors · 2025-11-24
>
> We sincerely appreciate the reviewer’s time and feedback. We have updated the paper with further clarifications and discussion (highlighted in blue). Below, we address each question individually:
>
> **Q1. Describe the method in more detail in Fig1/Abstract/Intro**
>
> A: Thanks for your suggestion. We have added more detail in the Abstract and Intro in the revised paper.
>
> **Q2. Investigate the impact of the sliding window size on LV-Eval and InfiniteBench**
>
> A: Thank you for the suggestion. We compare the AHN-augmented Qwen2.5-7B with a sliding window attention (SWA) baseline on both LV-Eval and InfiniteBench across a wide range of window sizes. Both methods use 128-token attention sinks, and we sweep sliding window sizes from 1k to 96k on the 128k-context sets of the two benchmarks. As shown in Figure 4, the AHN-augmented model maintains competitive performance across all window configurations, and consistently outperforms sliding window attention (SWA). Notably, as the inference window size increases from 1k to 16k, the performance improves steadily, highlighting the importance of a large attention window for extra-long context tasks. Beyond this range, however, we observe a noticeable performance drop, after 64k on LV-Eval and 96k on InfiniteBench, which may be attributed to the attention-dilution effect, where the attention distribution becomes overly diffuse when the number of keys grows very large, weakening the model’s ability to focus on relevant information. Balancing performance and computational cost, we therefore adopt a 32k sliding window as the default configuration for inference. We have included these discussions in the revised paper.
>
> **Q3. Performance on short-context tasks, such as MMLU(-PRO) or GSM8k or GPQA**
>
> A: Thank you for your comments. Since our motivation is to address the efficiency bottleneck of attention in extra-long-context scenarios (attention has no efficiency problem on short-context tasks), AHNs are designed to activate only when the sequence length exceeds a large sliding-window size (e.g., 32k).
> For the short-context tasks, MMLU, MMLU-Pro, GSM8k and GPQA, their average question lengths are only 10.1 words, 57.4 tokens, 46.9 words and 169 tokens. Their sequence lengths are far below the window size of 32k for inference, so AHNs do not activate, and the model is identical to the base full-attention model. Therefore, the AHN-augmented model achieves identical performance to the base attention model on these short-context benchmarks.
>
> **Q4. Implementation details**
>
> A: Thanks for your comments. AHNs do not introduce separate QKV projection layers, aligning with our motivation to transform the lossless memory (i.e., the KV cache) from attention into a fixed-size compact memory. The “extra param” in Table 2 has the same meaning as the unfrozen parameters of AHNs. Taking AHN-GDN as an example, only the parameters involved in Equations 5 and 6 are learnable. For each attention head, these trainable parameters consist of the gating weights $W_\alpha \in \mathbb{R}^{D \times 1}$ in $\alpha(\cdot)$, $W_\beta \in \mathbb{R}^{D \times 1}$ in $\beta(\cdot)$, $W_\gamma \in \mathbb{R}^{D \times 1}$ in $\gamma(\cdot)$ as well as the output projection $W_o \in \mathbb{R}^{H \times H}$. Here,  $D$ and $H$ denote the hidden dimension and the head dimension, respectively. With $N_\text{q}$ attention heads, the model contains $N_\text{q}$ such sets of parameters, amounting to only $\sim 0.4\%$ parameters relative to the frozen base model. Thanks for your suggestion, we have included these details in the revised paper.

---

> ### Author Response · Authors · 2025-11-24
>
> **Q5. Comparison to MiL (The Mamba in the Llama: https://arxiv.org/abs/2408.15237), LoLCATs (https://arxiv.org/abs/2410.10254) and HQLT (https://arxiv.org/abs/2506.00744)**
>
> A: Thank you for your comments. Regarding HQLT (NeurIPS 2025), we would like to clarify that our work was developed independently. The core idea, initial implementation, and draft of our method were completed before HQLT appeared online, and our submission is not derived from or influenced by it. We can provide timestamped evidence to the ACs if needed.
>
> Below we summarize the main differences between AHNs and MiL, LoLCATs, and HQLT:
>
> **Model architecture**
>
> Different from MiL, which distills the entire attention layer into a linear RNN, AHN only activates when the sequence length exceeds a large sliding window. Different from the small attention window of 64 used in HQLT and LoLCATs, AHN adopts a much larger 32k sliding-window size during inference. Since quadratic attention remains efficient for short and medium sequences, the quadratic-complexity bottleneck only appears when sequences become extra long. This motivates us to set a substantially larger attention window so that AHNs activate only when the sequence length exceeds this window and attention begins to encounter efficiency issues.
>
> **Target Tasks and Evaluation Setting**
>
> MiL, LoLCATs, and HQLT evaluate models mainly on short-context tasks (e.g., ARC and HellaSwag with sequence length <128), where linear RNNs cannot demonstrate efficiency advantages. Besides lacking efficiency gains, these methods must also make additional efforts to match the performance of attention on these short-context tasks.
>
> In contrast, AHN targets extra-long-context tasks (e.g., LV-Eval and InfiniteBench with 128k sequence length). AHN does not activate when the sequence length is shorter than the 32k window size, so the model operates exactly as a standard Transformer. In other words, the performance of our method on short-context tasks is identical to full attention, and no extra effort is required to match attention’s performance. For extra-long-context tasks such as LV-Eval and InfiniteBench, AHN not only achieves performance comparable to full attention, but also significantly reduces FLOPs and memory cache.
>
> **Training method**
>
> Compared with MiL and HQLT, AHN’s training method is more efficient. MiL needs to train the token mixer parameters with 20B tokens, and HQLT trains the entire model from scratch with 15B tokens. In contrast, we freeze the base model’s all parameters and only train the newly added AHN parameters (about 0.4% relative to the base model) with only 1B tokens. Compared to LoLCATs, which trains models through multiple stages of Attention Transfer and Low-rank Linearizing, AHN uses a simple one-stage self-distillation process.
>
> **In summary**, compared with these works, the goal of the AHN memory framework is to leverage the efficiency of RNNs specifically to address the computational bottleneck of attention on extra-long sequences. The distinct insight introduced by AHNs is to employ a large sliding-window size (eg, 32k) for attention, such that the RNN-like AHN modules only activate when the sequence length exceeds this window. This design provides two major advantages:
>
> 1) It highlights the efficiency benefits of RNNs on extra-long context tasks (e.g., 128k), achieving substantial FLOP and memory-cache savings. In contrast, for short-context tasks where attention is already efficient, introducing RNNs provides no efficiency gain.
>
> 2) It requires no additional effort to preserve attention performance on short-context tasks, because AHNs remain inactive in these regimes, and the model behaves exactly as a pure attention-based Transformer.
>
> Thanks for your suggestion. We have included these discussions to the revised paper.
>
> **Q6. Double check whether Figure 1 can be placed before the abstract**
>
> A: Yes, it is ok to place Figure 1 before the abstract, like these ICLR 2025 papers:
> https://proceedings.iclr.cc/paper_files/paper/2025/file/cf3d7d8e79703fe947deffb587a83639-Paper-Conference.pdf
>
> https://proceedings.iclr.cc/paper_files/paper/2025/file/bb7b958d2c11766ddcbc53332f65dab8-Paper-Conference.pdf
>
> https://proceedings.iclr.cc/paper_files/paper/2025/file/ea0b28cbbd0cbc45ec4ac38e92da9cb2-Paper-Conference.pdf
>
> **Q7. Move the related work to the end**
>
> A: Thanks for your suggestion. We have moved the related works to the end of the paper.

---

> ### Author Response · Authors · 2025-11-24
>
> **Q8. What sliding window sizes were used? What is meant by randomizing the starting position?**
>
> A: Thanks for your comments.
>
> Lossless memory size = attention sinks tokens (starting position) + sliding window tokens.
>
> We treat attention sinks as uncompressed “starting-position” for the AHNs.  For each training sample, the total lossless memory size is uniformly sampled from [32, 64, 128, 256, 512, 1024, 2048, 4096, 8192] after filtering out values smaller than one-eighth of the sample sequence length. Then, we choose the attention-sink size by uniformly sampling from [0, 32, 64, 128, 512, 2048, 4096] after removing candidates larger than half of the sequence length. Thus, during training, the sliding window size is randomly sampled up to 8,192, and the maximum sequence length is 24k. Thanks for your suggestion. We have incorporated these details in the revised paper.
>
> **Q9. About Figure 5**
>
> A: Thanks for your comments. Figure 5 shows an example from AceMath-Instruct-Training-Data. The full sequence has 811 tokens, and we evaluate the AHN-augmented model using a 512-token sliding window (AHN activates once the context surpasses 512 tokens). The snippet shows the gradients for the first 139 tokens. Thanks for your suggestion, we have included these details in the revised paper.

---

### Official Review · Reviewer_C5cT · 2025-11-04

**Soundness:** 3
**Presentation:** 2
**Contribution:** 2
**Rating:** 2
**Confidence:** 5

**Summary:**

This paper introduces Artificial Hippocampus Networks (AHNs), a novel and efficient framework for long context modeling in Transformers. The work addresses the fundamental trade off between the high fidelity, growing memory of attention's KV cache and the efficient, fixed size memory of Recurrent Neural Network like models. Inspired by the Multi Store Model of memory from cognitive science, the proposed method uses a hybrid approach. It maintains a sliding window of the KV cache as a lossless short term memory, while the AHN module, a learnable component, recurrently compresses KV pairs that exit the window into a fixed size long term memory state. This AHN module can be implemented using various modern RNN like architectures, such as Mamba2 or GatedDeltaNet. A key contribution is the efficient training method. Instead of training from scratch, the authors use a parameter efficient self distillation framework where the AHN parameters are trained to match the output distributions of a frozen, full attention "teacher" model. Experiments, particularly on the Qwen2.5 3B model, show that this method dramatically reduces computational and memory costs (e.g., 40.5% fewer FLOPs and a 74.0% smaller memory cache on LV Eval) while simultaneously and surprisingly improving average scores on long context benchmarks.

**Strengths:**

1. Novel Hybrid Framework: The paper introduces a novel hybrid memory framework, drawing inspiration from cognitive science to blend lossless, sliding window attention with a fixed size compressive memory. This two part memory system is a creative approach to the long context problem. The concept is presented clearly, with diagrams that effectively illustrate the mechanism.

2. Efficiency and Training Method: The primary practical strength is the method's efficiency. By design, it achieves constant memory cache size and linear computational complexity, which is a significant improvement over standard attention. The paper demonstrates this leads to substantial reductions in FLOPs (40.5%) and memory cache (74.0%) on a 3B parameter model. The self distillation training approach is also a notable contribution, as it allows these gains to be achieved by efficiently adapting existing pretrained models.

**Weaknesses:**

1. Lossy Compression and Exact Recall: The most significant weakness is the model's performance on exact recall tasks. The RULER benchmark's needle in a haystack (NIAH) tasks (Table 5) show that the AHN augmented model fails at retrieving specific facts from deep context, performing just as poorly as a simple sliding window and far worse than full attention. This is a critical trade off. The main paper's strong results on LV Eval and LongBench may be because these benchmarks test for general understanding rather than high fidelity recall. This limitation makes the method unsuitable for tasks that depend on precise, long range fact retrieval.

2. Unclear Comparison to Full Attention: The paper presents the very surprising result that the AHN model outperforms the full attention baseline (e.g., 5.88 vs 4.41 on LV Eval). This is counterintuitive, as the full attention model has access to strictly more information. The paper does not adequately explain this. Figure 3c shows the baseline model's perplexity spiking after 32K tokens, which suggests the baseline Qwen2.5 3B model was not pretrained for such long contexts. If this is the case, the comparison is not entirely fair. The performance gain might not come from the superiority of the AHN architecture, but rather from the self distillation process itself acting as an effective form of context length extension training, which the baseline did not receive.

3. Focus on Post Training Adaptation: The paper exclusively validates AHNs as a module added after pretraining, using a parameter efficient, frozen backbone approach. This leaves open the question of how an architecture with AHNs would perform if pretrained from scratch. The current approach's performance is inherently capped by the teacher model's capabilities. Demonstrating that this memory framework is stable and effective during full pretraining would make a much stronger case for it as a new architectural paradigm, rather than just an efficient adaptation technique.

**Questions:**

1. Regarding the Full Attention Comparison: Could you please clarify the extraordinary result where the AHN model (5.88) significantly outperforms the full attention baseline (4.41) on LV Eval? As noted in my weaknesses, Figure 3c suggests the baseline model is not functional beyond its 32K context window. Is the baseline model simply failing due to a lack of long context pretraining? And is it possible that the self distillation process itself is the primary source of the performance gain, by effectively fine tuning the model for longer contexts, rather than the AHN's compressive memory?

2. Reconciling Compression and Recall Failure: The gradient visualization in Figure 5 suggests the AHN learns to selectively compress information, preserving important symbols and numbers. However, the appendix (Table 5) shows a total failure on needle in a haystack tasks. How do you reconcile these two findings?

3. Training Practicality: The paper emphasizes the efficiency of the self distillation training. Could you provide more concrete details on the training cost? For example, how many GPU hours or what fraction of the original pretraining cost does this self distillation phase require? Furthermore, do you have any preliminary results or hypotheses on the feasibility and stability of training a model with AHNs from scratch?

---

> ### Author Response · Authors · 2025-11-24
>
> We sincerely appreciate the reviewer’s time and feedback. We have updated the paper with further clarifications and discussion (highlighted in blue). Below, we address each question individually:
>
> **Q1: Model performance on the RULER benchmark's Needle-In-A-Haystack (NIAH) tasks**
>
> A: Thank you for your comments. Our work primarily targets the efficiency challenges of extra-long sequence modeling for general understanding tasks. Compared with full attention, AHNs substantially reduce both FLOPs and memory usage. However, there is no free lunch. Because AHNs compress all information outside the sliding window into a fixed-size recurrent memory, they are not designed for tasks requiring exact token-level recall. Therefore, it is expected that AHNs may underperform on Needle-In-A-Haystack (NIAH) tasks, where the “needle’’ appears as an isolated token sequence that is semantically unrelated to its surrounding context. In such settings, retrieval-augmented methods (e.g., RAG) are more appropriate for achieving precise recall.
>
> **Q2: Unclear Comparison to Full Attention.**
>
> A: Thanks for your comments. According to the configurations of Qwen2.5, the original `max_position_embeddings` for 3B, 7B, 14B model are 32k, 131k and 131k, respectively. We believe the comparison is fair  because the maximum training sequence length for AHNs is only 24k. Although AHNs are not trained on the sequence length of 128k, a key advantage of AHNs is their ability to generalize smoothly to much longer contexts than those seen in training. For very long sequences (e.g., 128k), the full-attention baseline may underperform due to attention dilution, where the the attention distribution becomes overly diffuse when the number of keys grows very large, weakening the model’s ability to focus on relevant information. In contrast, AHNs mitigate attention dilution by combining a large sliding-window attention (SWA) with a fixed-size compressed memory that aggregates information outside the local window. This design allows AHNs to maintain focused attention while scaling efficiently to extra-long contexts.
>
> **Q3: Focus on Post Training Adaptation (How about training a model with AHNs from scratch?)**
>
> A: Thank you for your comments. AHNs employ a large sliding-window size (default 32k) during inference. When the sequence length is below 32k, AHNs do not activate, and the model behaves identically to the full-attention baseline. Therefore, we believe it is not necessary to pretrain an AHN-augmented model from scratch. The pretraining of the full-attention model naturally serves as the pretraining stage for AHN-augmented models.
>
> That said, exploring continual, full-model fine-tuning of AHN-augmented models may be worthwhile. The challenge, however, lies in accessing the same large-scale training data used for the base LLMs. Since AHNs remain inactive for sequences shorter than 32k tokens, full-model fine-tuning would require training on the same data distribution to preserve the base model’s performance on short- and medium-context tasks.
>
> In contrast, our current efficient self-distillation approach freezes the entire base LLM and only trains the AHN parameters. This completely avoids the need to maintain short-context performance and requires only 1B tokens from the open-source ChatQA-2 dataset to train AHNs. This lightweight training pipeline makes the method easy to open-source and fully reproducible, enabling the research community to reproduce our results without relying on massive proprietary datasets or large-scale computing resources.
>
> **Q4: Reconciling Compression and Recall Failure.**
>
> A: Thanks for your comments. These two observations are not in conflict. Figure 5 shows that AHNs can selectively store informative tokens: the AHN module preserves more information for salient tokens, e.g., key numbers or operators, and less for uninformative context. This indicates that the compressive memory is not uniformly lossy, but instead prioritizes high-value tokens.
>
> In contrast, Needle-In-A-Haystack (NIAH) requires perfect recall of an arbitrary token sequence, where even a minor compression error causes failure. In Table 5, AHNs use a 32k sliding window while the total sequence length is 128k. It is inherently difficult for a fixed-size memory to store all 96k off-window tokens with exact fidelity. Moreover, in NIAH, the inserted “needle” number is completely unrelated to the surrounding fiction text, making it extremely difficult for AHNs to recognize that this synthetic number should be prioritized and preserved.
>
> In short, AHNs can select important tokens, but they are not designed to store all tokens with exact precision. Therefore, the behaviors shown in Figure 5 and Table 5 are fully consistent.

---

> ### Author Response · Authors · 2025-11-24
>
> **Q5: Training Practicality**
>
> A: Thanks for the comments. Our self-distillation procedure is highly lightweight because the entire base LLM is frozen and only the newly added AHN parameters (about 0.4% relative to the base model size) are optimized. AHNs are trained on the open-source ChatQA-2 dataset for one epoch (1B tokens), which is only 0.006% of the 18T tokens used to pre-train Qwen2.5.
>
> For the Qwen2.5-7B model, training the AHN module takes only 10 hours on 32 A100 GPUs with a batch size of 128, corresponding to approximately 740 update steps. This demonstrates the strong practicality and efficiency of our training pipeline. Thanks for your suggestion. We have included these discussions and details in the revised paper.

---

### Official Review · Reviewer_EXLV · 2025-11-08

**Soundness:** 3
**Presentation:** 2
**Contribution:** 2
**Rating:** 4
**Confidence:** 4

**Summary:**

This paper proposed a framework, named artificial hippocampus network (AHN), which integrates sliding window attention with recurrent memory. Concretely, sliding window attention with fixed attention context handles short-term memory, and recurrent memory, implemented with linear recurrent networks, compresses and maintains long-term contextual information. Rather than training from scratch with the standard cross entropy loss, the authors leverages self-distillation by minimizing the KL-divergence between a teacher model and the memory-augmented model.

Experiments were conducted on top of QWen-2.5-Instruct models, with ChatQA2 as training data. Evaluation were performed on long-context benchmarks, including LV-Evan, InfiniteBench and Ruler.

**Strengths:**

The AHN framework is well-motivated to integrate SWA and recurrent-memory.

**Weaknesses:**

There are several weaknesses of this paper:

1. The proposed AHN framework is very similar to previous recurrent-memory studies, such as infiniti-attention [1]. The major difference is that infiniti-atention is using chunk-wise attention and update the recurrent memory in a chunk-wise way. However, this paper does not make any comparison to these related work.

2. There are no reported numbers on efficiency of model training after adding the recurrent memory module, comparing with purely sliding window attention and/or full attention.

3. The improvements of the proposed models upon pure SWA baseline are not impressiveon LongBench tasks. On Ruler, the scores of needle-in-haystack tasks are still near random baselines.


References:

1. Leave No Context Behind: Efficient Infinite Context Transformers with Infini-attention

**Questions:**

Questions:

1. Does the recurrent memory shares the same set of $W_Q, W_K, W_V$ with SWA? If yes, the only additional parameters in the recurrent memory module are from the backend linear recurrent network, such as Mamba, DeltaNet or GatedDeltaNet?

2. What is the sequence length and the attention context window during training?

---

> ### Author Response · Authors · 2025-11-24
>
> We sincerely appreciate the reviewer’s time and feedback. We have updated the paper with further clarifications and discussion (highlighted in blue). Below, we address each question individually:
>
>
> **Q1: Discuss the difference between Infinite-Attention and AHN**
>
> A: Thank you for your comments. The two approaches differ in several key aspects:
>
> **Memory update mechanism.**
>
> As you pointed out, Infinite-Attention performs chunk-wise attention and updates its recurrent memory in a chunk-wise way. In contrast, AHN is built on a standard decoder-only autoregressive Transformer and updates its compressed memory in a token-wise manner. This token-wise design allows AHN to be integrated seamlessly into existing popular base models, and it enables flexible configuration of the sliding-window attention (SWA) size according to available hardware memory.
>
> **Model architecture.**
>
> Infinite-Attention relies on chunk-wise attention with a relatively small fixed segment size (window size), ie 2048. In contrast, AHN adopts a much larger 32k sliding-window size during inference. Since quadratic attention remains efficient for short and medium sequences, the quadratic-complexity bottleneck only appears when sequences become extra long. This motivates us to set a substantially larger attention window so that AHNs activate only when the sequence length exceeds this window and attention begins to encounter efficiency issues.
> This design brings a key advantage: It requires no additional effort to preserve attention performance on short-context tasks, because AHNs remain inactive in these regimes, and the model behaves exactly as a pure attention-based Transformer.
>
> **Training method.**
>
> Infini-attention does not disclose its overall training cost; it only reports training for 30K steps with a batch size of 64 before fine-tuning on the passkey retrieval task.
> In contrast, our paper proposes an efficient self-distillation strategy for training AHNs. AHN requires only 740 steps with a batch size of 128**, using only 1B tokens. This highlights the simplicity and data efficiency of our method.
>
> **Openness and reproducibility.**
>
> To the best of our knowledge, Infinite-Attention remains closed-source and lacks reliable third-party implementations. In contrast, we commit to fully open-sourcing all AHN code and models, including both training and evaluation pipelines, to ensure transparency and reproducibility for the community.
>
> Thank you for your suggestion. We have incorporated these discussions into the revised paper.
>
> **Q2: Numbers on efficiency of model training after adding the recurrent memory module**
>
> Thanks for your comments. Our method uses a self-distillation training strategy in which only the AHN parameters are optimized, while all parameters of the base LLM remain frozen. For Qwen2.5-7B model, training its AHN only takes 10 hours on 32 A100 GPUs, using just 1B tokens from ChatQA 2 for one epoch. Although the maximum training sequence length is 24k, the resulting model generalizes to much longer sequences (e.g., 128k) during inference. Importantly, our method does not require re-training the base LLM.
>
> Per the reviewer’s request to compare the training efficiency of sliding-window attention, AHN-augmented models, and full attention, we calculate their training FLOPs under a unified setting: AdamW optimizer, next-token prediction, full-model training, batch size 128, 24k sequence length, and 8k sliding window size. The FLOPs per training step are summarized as follows:
>
> Table: One-step training FLOPs (1e17) under the setting of AdamW optimizer, next-token prediction, full-parameter tuning, batch size 128, sequence length 24k, and sliding-window size 8k.
>
> | Method                 | 3B     | 7B     | 14B    |
> |-----------------------|--------|--------|--------|
> | Full attention        | 0.6348 | 1.1519 | 2.5396 |
> | Attention Sinks + SWA | 0.5405 | 1.0334 | 2.2252 |
> | AHN-GDN               | 0.5422 | 1.0359 | 2.2319 |
>
>
> Thank you for your suggestion. We have incorporated these discussions into the revised paper.

---

> ### Author Response · Authors · 2025-11-24
>
> **Q3: Performance on LongBench and Ruler**
>
> A: Thanks for your comments.
>
> **LongBench.** The tasks in LongBench are actually not the target scenario for AHNs, as their average sequence lengths are relatively short (approximately 1k–22k), which is below the default 32k sliding-window size used by AHNs during inference. To evaluate AHNs on LongBench, we have to reduce the SWA size to 8k. However, this makes the performance gains over the sliding-window baseline less pronounced compared to benchmarks with truly long contexts such as LV-Eval and InfiniteBench (128k sequence length).
>
> **Ruler.** First, note that 25% is not the random baseline performance. In addition, AHNs are not fine-tuned on Needle-In-A-Haystack (NIAH) tasks, so the reported result reflects zero-shot accuracy. In fact, the performance of AHNs on Ruler’s NIAH tasks aligns with our expectations. The core motivation of AHNs is to alleviate the quadratic-complexity bottleneck of attention on extra-long sequences, rather than to perfectly preserve every token outside the attention window. As such, we do not expect AHNs to match full-attention models on all tasks. There is no free lunch. Since AHNs compress all information outside the sliding window into a fixed-size memory, they are not ideal for tasks requiring precise long-range retrieval, such as NIAH, especially when the “needle” is semantically unrelated to the surrounding context; in such cases AHNs may naturally deem it unimportant and not preserve it in the compressed memory. For tasks that demand exact long-context recall, retrieval-augmented methods (RAG) may be a more suitable solution.
>
> **Q4: Does the recurrent memory share the same set of SWA?**
>
> A: Yes, the recurrent memory shares the same set of SWA, aligning our motivation to convert the lossless memory (KV cache) into compact fixed memory.  Thus, the added AHN parameters only include the input gate, forget gate, output gate, and output projection of RNN-like architectures. For output projection, we use group linear layer (with the number of groups equal to the number of attention heads) to further reduce the parameters. Therefore, only ~0.4% parameters are introduced and trained, highlighting the efficiency of our approach. Thank you for your suggestion. We have incorporated these discussions into the revised paper.
>
> **Q5: What is the sequence length and the attention context window during training?**
>
> A: During training, we use a maximum sequence length of 24k tokens. For each example, the attention-sink size is uniformly sampled from [0, 32, 64, 128, 512, 2048, 4096] after removing any candidates larger than half of the sequence length. The total token number of lossless memory (attention sinks + sliding window) is uniformly sampled from [32, 64, 128, 256, 512, 1024, 2048, 4096, 8192] after filtering out values smaller than one-eighth of the sequence length. The training random sizes of the attention sink and the sliding window improve model robustness and performance, as shown in Table 4 of the paper. Thank you for your suggestion. We have incorporated these discussions into the revised paper.

---

### Author Response · Authors · 2025-12-02
**Rebuttal Summary**

Dear Reviewers and ACs,

We sincerely thank all reviewers for their time and feedback, and we also appreciate the ACs for their continued support of our research community. We have carefully addressed the reviewers’ questions and clarified misunderstandings in the rebuttal. Besides, we have updated the paper with additional content highlighted in blue. For convenience, we summarize the key points below to assist the ACs in their evaluation.

**Q1: Compassion with Infinite-attention, MiL, LoLCATs and co HQLT** (from Reviewer EXLV, RgMK and YqVp)

A: We have provided detailed comparison and discussion in the rebuttal and have incorporated them into the revised paper. In summary, compared with these works, **the goal of the AHN memory framework** is to leverage the efficiency of RNNs specifically to address the computational bottleneck of attention on extra-long sequences. The **distinct insight** introduced by AHNs is to employ a large sliding-window size (eg, 32k) for attention, and RNN-like AHN modules only activate when the sequence length exceeds this window. This design provides two major advantages:

1) It **highlights the efficiency benefits of RNNs** on extra-long context tasks (e.g., 128k), achieving substantial FLOP and memory-cache savings. In contrast, for short-context tasks where attention is already efficient, introducing RNNs provides no efficiency gain.

2) It **requires no additional effort to preserve performance on short-context tasks**, because AHNs remain inactive in these regimes, and the model behaves exactly as a pure attention-based Transformer.


**Q2: Model performance on Needle-In-A-Haystack (NIAH) tasks** (from Reviewer EXLV and C5cT)

A: **AHNs target the efficiency challenges of general extra-long context tasks and substantially reduce both FLOPs and memory cache**. However, **there is no free lunch**. Because AHNs compress all information outside the sliding window into a fixed-size recurrent memory and **discard the KV cache outside the window**, they are not designed for tasks requiring exact token-level recall of information located far outside the window. Therefore, **it is expected** that AHNs may underperform on Needle-In-A-Haystack (NIAH) tasks, where the “needle’’ appears as an isolated token sequence that is semantically unrelated to its surrounding context. In such settings, retrieval-augmented methods (e.g., RAG) are more appropriate for achieving precise recall.


**Q3: Pretrain architecture with AHNs from scratch** (from Reviewer C5cT)

A: Our memory framework adopts large sliding window size for attention, and AHNs only activate when the sequence length exceeds the window. During pretraining, the maximum sequence length is short (e.g., 4k for Qwen2.5), so AHNs remain inactive. Therefore, **the pretraining stage of our memory framework is identical to that of the base full-attention model**.


**Q4. Performance on short-context tasks** (from Reviewer RgMK)

A: Since our motivation is to address the efficiency bottleneck of attention in extra-long-context scenarios (attention has no efficiency problem on short-context tasks), AHNs are designed to activate only when the sequence length exceeds a large sliding-window size (e.g., 32k). For short-context tasks, AHNs remain inactive, and the model behaves exactly as a standard full-attention model. Therefore, **the performance on short-context tasks is identical to that of the base full-attention model**.


**Q5. Additional model details and ablation experiments** (from Reviewers EXLV, C5cT, RgM,K and YqVp)

A: We have added further model details and ablation experiments in both the rebuttal and the revised paper.

---

### Meta-Review · Area_Chair_wJ2B · 2026-01-06

**Summary:**

The paper propose to add a RNN mechanism to attention to help with long context. They show some improvement in the score but worse results in recall. The tradeoff was studied well enough and also comparison to other RNN based baselines is lacking.

**Reviewer Concerns:**

The reviewers answered some of the concerns but I am not satisfied with their answer about the recall problem neither the comparison to RNN baselines. This is insufficient and require a major revision and a better explanation.

**Reviewer Scores:**

I don't think they will change their score in a significant manner that will make the paper acceptable.

---

### Decision · Program_Chairs · 2026-01-26

Reject